# Pannexin 1 and pannexin 3 differentially regulate the cancer cell properties of cutaneous squamous cell carcinoma

Brooke L. O'Donnell[1] , Danielle Johnston[1], Ayushi Bhatt[2], Zahra Kardan[3], Dan Stefan[1], Andrew Bysice[4], Samar Sayedyahossein[5], Lina Dagnino[5,6], Matthew Cecchini[7], Sampath Kumar Loganathan[3,8], Kathryn Roth[4,9] and Silvia Penuela[1,6]

[1]*Department of Anatomy and Cell Biology, Schulich School of Medicine and Dentistry, University of Western Ontario, London, Ontario, Canada*

[2]*Faculty of Medicine, Schulich School of Medicine and Dentistry, University of Western Ontario, London, Ontario, Canada*

[3]*Department of Surgical and Interventional Sciences, Faculty of Medicine and Health Sciences, McGill University, Montreal, Quebec, Canada*

[4]*Department of Otolaryngology – Head and Neck Surgery, Schulich School of Medicine and Dentistry, University of Western Ontario, London, Ontario, Canada*

[5]*Department of Physiology and Pharmacology, Schulich School of Medicine and Dentistry, University of Western Ontario, London, Ontario, Canada*

[6]*Division of Experimental Oncology, Department of Oncology, Schulich School of Medicine and Dentistry, University of Western Ontario, London, Ontario, Canada*

[7]*Department of Pathology and Laboratory Medicine, Schulich School of Medicine and Dentistry, University of Western Ontario, London, Ontario, Canada*

[8]*Departments of Otolaryngology – Head and Neck Surgery, Biochemistry and Experimental Medicine, Rosalind and Moris Goodman Cancer Research Institute, Faculty of Medicine and Health Sciences, McGill University, Montreal, Quebec, Canada*

[9]*London Regional Cancer Program, London Health Sciences Centre, London, Ontario, Canada*

Handling Editors: Peying Fong & Jorge Contreras

The peer review history is available in the Supporting Information section of this article (https://doi.org/10.1113/JP286172#support-information-section).

**Abstract figure legend** PANX1 and PANX3 show opposite expression patterns in patient-derived normal skin and cutaneous squamous cell carcinoma (cSCC) tumours, with PANX1 increased and *PANX3* mRNA decreased in cSCC tumours compared to skin. Within the tumour PANX1 localizes to all regions, including tumour nests which house cSCC cancer cells and tumour stroma which contains tumour-infiltrating lymphocytes, cancer-associated fibroblasts and blood vessels. *PANX1* deletion by CRISPR/Cas9 editing and PANX1 channel inhibition via probenecid (PBN) and spironolactone (SPL) reduces SCC-13 cancer cell properties of SCC-13 such as growth and migration. Conversely, global *Panx3* (knockout) KO mice subjected to the 7,12-dimethylbenz(a)anthracene/12-otetradecanoylphorbol-13-acetate (DMBA/TPA) cutaneous carcinogenesis model have increased papilloma growth and incidence. Created with Biorender.com.

This article was first published as a preprint. O'Donnell BL, Johnston D, Bhatt A, Kardan Z, Stefan D, Bysice A, Sayedyahossein S, Dagnino L, Cecchini M, Loganathan SK, Roth K, Penuela S. 2024. Pannexin 1 and Pannexin 3 differentially regulate the tumorigenic properties of cutaneous squamous cell carcinoma. bioRxiv. https://doi.org/10.1101/2024.04.08.588550

The Journal of Physiology

**Abstract**   Pannexin (PANX) channels are present in skin and facilitate the movement of signalling molecules during cellular communication. PANX1 and PANX3 function in skin homeostasis and keratinocyte differentiation but were previously reduced in a small cohort of human cutaneous squamous cell carcinoma (cSCC) tumours compared to normal epidermis. In our study we used SCC-13 cells, limited publicly available RNA-seq data and a larger cohort of cSCC patient-matched samples to analyse PANX1 and PANX3 expression and determine the association between their dysregulation and the malignant properties of cSCC. In a bioinformatics analysis, *PANX1* transcripts were increased in cSCC and head and neck SCC tumours compared to normal tissues, but *PANX3* mRNA showed no differences. However, in our own cohort *PANX3* transcripts were decreased in cSCC compared to patient-matched aged skin, whereas PANX1 protein was upregulated in cSCC. PANX1 localized to all regions within the cSCC tumour microenvironment, and increased levels were associated with larger tumour dimensions. To investigate PANX1 function in SCC-13 cells, we deleted *PANX1* via CRISPR/Cas9 and treated with PANX1 inhibitors, which markedly reduced cell growth and migration. To assess PANX3 function in cutaneous carcinogenesis, we employed the 7,12-dimethylbenz(a)anthracene/12-otetradecanoylphorbol-13-acetate (DMBA/TPA) model using our global *Panx3* knockout (KO) mice, where 60% of wild-type and 100% of KO mice formed precancerous papillomas. Average papilloma volumes at endpoint were significantly increased in KO mice and showed moderate evidence of increases in KO mice over time. Collectively, these findings suggest PANX1 and PANX3 dysregulation may have potential tumour-promoting and tumour-suppressive effects for keratinocyte transformation, respectively.

(Received 12 March 2024; accepted after revision 23 October 2024; first published online 18 November 2024)
**Corresponding author** S. Penuela, Department of Anatomy and Cell Biology, Schulich School of Medicine and Dentistry, University of Western Ontario, Room 00061B Dental Science Building, London, ON N6A5C1, Canada. Email: silvia.penuela@schulich.uwo.ca

## Key points

- Pannexin 1 (PANX1) and pannexin 3 (PANX3) are channel-forming proteins which are critical in the normal maintenance and function of keratinocytes in the skin but may become altered in cutaneous squamous cell carcinoma (cSCC) tumours.
- In this study we used a combination of culture models, mouse models and patient-derived tissues. We found PANX1 levels are increased in cSCC tumours and present in all tumour regions, functioning to promote cSCC cell growth and migration.
- Conversely, PANX3 levels are decreased in cSCC tumours, and this protein reduces the incidence and growth of precancerous lesions.
- Taken together our data indicate that in cSCC these pannexin family members seem to have opposite effects, in either promoting or restricting cancer cell properties.
- These results help us to better understand the mechanisms of malignant transformation of keratinocytes and offer a new potential therapeutic target for the treatment of advanced cSCC.

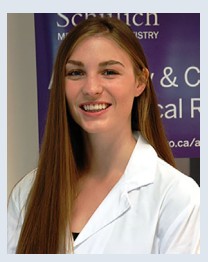

**Brooke L. O'Donnell** graduated with a BMSc (Hons) in physiology at the University of Western Ontario and received her PhD in anatomy and cell biology at Western under the supervision of Dr Silvia Penuela. Her PhD research focused on the function of the channel proteins pannexin 1 and pannexin 3 and their isoforms in skin health and ageing as well as their dysregulation in keratinocytic skin cancers. She is currently a research associate at the University of Virginia in Dr Brant Isakson's laboratory where she is investigating pannexins and the physiological consequences of their posttranslational modifications in the vasculature.

## Introduction

Cutaneous squamous cell carcinoma (cSCC) is the second most common form of skin cancer arising from squamous keratinocytes in the upper layers of the epidermis or stem cell populations in cutaneous glands and hair follicles (Firnhaber, 2012; Kallini et al., 2015). Despite accounting for only 20% of non-melanoma skin cancer (NMSC) cases (Firnhaber, 2012; Madan et al., 2010), cSCC is responsible for most NMSC-related deaths often due to tumour metastasis (Nissinen et al., 2016), and the incidence of cSCC is projected to continue increasing worldwide as the population ages (Lomas et al., 2012; Neagu et al., 2019). Generally, surgical resection is a viable treatment option for small and localized tumours, but approximately 5% of cSCC progress to advanced disease, becoming highly invasive (Schmults et al., 2013). This contributes to a very high recurrence rate and requires additional treatment methods such as chemotherapy, radiation, targeted therapies and immunotherapy, which have lower success rates that vary depending on the patient (Aboul-Fettouh et al., 2021; Fahradyan et al., 2017). Despite these considerations the mechanisms of keratinocyte malignant transformation into cSCC are still poorly understood (Ratushny et al., 2012).

Within the skin, cells express pannexins (PANX1, PANX2, PANX3) which are canonically known for their roles as channel-forming glycoproteins (Panchin et al., 2000) that facilitate cellular communication by metabolite and ion conductance (Bao et al., 2004; Ishikawa et al., 2011; Iwamoto et al., 2010; O'Donnell & Penuela, 2023). PANX1 has been demonstrated to play a role in murine skin development and the establishment of proper skin architecture, with protein levels being high in young skin and decreasing with age in the absence of challenges such as wound repair (Penuela et al., 2014). Conversely, PANX3 is undetectable in newborn mouse dorsal skin but becomes upregulated with age and is highly abundant in aged mouse skin, functioning to maintain skin homeostasis during the process of ageing (O'Donnell et al., 2023). Both PANX1 and PANX3 have been localized to all living layers of keratinocytes, where they act in keratinocyte differentiation, adhesion and/or inflammation. During calcium-induced keratinocyte differentiation PANX1 levels decrease (Penuela et al., 2014), whereas PANX3 levels were found to either be induced or remain unchanged (O'Donnell et al., 2023; Zhang et al., 2019; Zhang et al., 2021), indicating distinct expression patterns. Interestingly, PANX1 localization changes from a cell membrane profile in basal, undifferentiated keratinocytes to a more intracellular profile in supra-basal, differentiated keratinocytes (Penuela et al., 2014), whereas PANX3 exhibits only cytoplasmic localization in both undifferentiated and differentiated cells (Cowan et al., 2012). Additionally,

overexpression of each protein in keratinocytic cell lines reduced proliferation, but only PANX1 overexpression increased keratin 14 levels (basal keratinocyte marker) and disrupted terminal differentiation, which normally facilitates the formation of epidermal architecture (Celetti et al., 2010). However, little is known about PANX expression and function in keratinocytic cancers where the normal keratinocyte differentiation programme is disrupted.

PANX1 has previously been implicated in melanoma tumorigenesis. Our group showed PANX1 levels are increased in melanoma cells compared to normal melanocytes, and *PANX1* mRNA is upregulated in patient melanoma tumour samples compared to control skin, with levels remaining high throughout all stages of melanoma disease progression (Freeman et al., 2019; Penuela et al., 2012). In human A375-P and A375-MA2 melanoma cells, decreased cellular growth rate, migration, invasion and mitochondrial metabolic activity, as well as increased melanin production, were observed when PANX1 channel activity was inhibited or PANX1 levels were reduced via shRNA knockdown. The malignant melanoma marker $\beta$-catenin, which was found to bind directly to the C-terminal tail of PANX1, was also significantly decreased (Freeman et al., 2019; Sayedyahossein et al., 2021). Altogether, this suggests a role for PANX1 throughout melanoma carcinogenesis, but little is known about PANX1 or PANX3 in NMSCs.

In a study using the 7,12-dimethylbenz(a)anthracene/12-otetradecanoylphorbol-13-acetate (DMBA/TPA) mouse cutaneous carcinoma model, which mimics many aspects of human cSCC (Abel et al., 2009), *Panx3* was found genetically linked to body mass index and tumorigenesis by quantitative trait loci analysis in male but not female mice. *Panx3* transcript levels were markedly reduced in papillomas and carcinomas compared to untreated tail skin (Halliwill et al., 2016). In human cSCC patient-derived tumours and aged epidermis, immunofluorescence microscopy analysis using a small cohort of samples indicated that both PANX1 and PANX3 were visibly reduced in cSCC tumours (Cowan et al., 2012). However, data from the Human Protein Atlas show conflicting results for PANX1 (PANX3 is not annotated), reporting that PANX1 is present at moderate levels in immunohistochemically stained cSCC tumour cores, where cells showed both membranous and cytoplasmic PANX1 localization. Collectively, these results suggest PANX1 and PANX3 levels are altered in cSCC tumorigenesis, but further investigation is needed to navigate the conflicting results and understand the effects of PANX dysregulation.

In this study we utilized the SCC-13 cell line, publicly available cSCC and head and neck SCC (HNSCC) patient expression data, patient-derived skin and cSCC tissues and a mouse model of cutaneous carcinogenesis

to evaluate PANX1 and PANX3 expression in cSCC compared to normal skin and their role in cSCC malignancy. In our patient sample cohort, we found that PANX1 protein is upregulated whereas *PANX3* mRNA is downregulated in human cSCC tumour fragments compared to patient-matched normal skin. Within SCC-13 cells, PANX1 exhibits both an intracellular and cell surface localization, whereas PANX3 is found predominantly in the cytosol. We also demonstrated that PANX1 promotes cancer cell properties such as growth and migration through its channel activity and is present in all regions within the cSCC tumour microenvironment and adjacent skin. On the contrary, PANX3 plays a tumour-suppressive role where it reduces the incidence and growth of precancerous papillomas.

## Methods

### Ethical approval

Animal experiments were approved by the Animal Care Committee at the University of Western Ontario (London, ON, Canada, Protocol 2022-025). At end-point mice were killed using $CO_2$ inhalation (in an increasing concentration). For collection of human samples the study was conducted in accordance with the latest version of the Declaration of Helsinki (except for registration in a database), and the protocol was approved by the Health Science Research Ethics Board (HSREB) of Western University and London Health Sciences Centre (London, ON, Canada, HSREB103381). Subjects provided informed consent to participate in this study, and tumour identity was determined by a pathologist.

### Cell culture

SCC-13 cells (RRID:CVCL_4029) were cultured in Dulbecco's modified Eagle's medium (DMEM, Gibco 12430062, Grand Island, NY, USA) supplemented with 10% fetal bovine serum (WISENT 098-150, Saint-Jean-Baptiste, QC, Canada). U-2 OS (ATCC HTB-96, Manassas, VA, USA; RRID:CVCL_0042), GBM17 (gifted from Dr Matthew Hebb, University of Western Ontario), Hs578T (ATCC HTB-126, RRID:CVCL_0332) and Hs578T *PANX1* knockout (KO) (Nouri-Nejad et al., 2021) cells used for positive and negative controls for western blotting were cultured in DMEM supplemented with 10% fetal bovine serum and 1% penicillin/streptomycin (Gibco 15140-122). Cells were maintained at 37°C and 5% $CO_2$ and dissociated using 0.25% trypsin/1 mM EDTA (WISENT 325-043-EL) for 7 min after one wash with 1× Dulbecco's phosphate buffered saline (DPBS) without $Ca^{2+}$ and $Mg^{2+}$ (Gibco 14190250) for SCC-13 cells or 3 min for all other cells. Cells were transfected using Lipofectamine 3000 (Invitrogen L3000015, Carlsbad, CA, USA) in Opti-MEM medium (Life Technologies 51985-034, Carlsbad, CA, USA) according to the manufacturer's protocol with 1 µg of PANX1 (InvivoGen puno1-hpanx1, San Diego, CA, USA) or PANX3 plasmid in a six-well plate. The PANX3 construct was modified from the original Origene plasmid (PS100001, Rockville, MD, USA) by NorClone Biotech Laboratories (London, ON, Canada) to replace the FLAG tag with an HA tag. Protein extracts were prepared 48 h after transfection for western blot positive controls.

### Protein isolation and western blotting

Protein lysates were prepared from cells using 1× radioimmunoprecipitation assay (RIPA) buffer, and from human tissue using 2× RIPA buffer after pulverizing in liquid nitrogen. Both extraction buffers also contained 1 mM sodium fluoride, 1 mM sodium orthovanadate and a half of a Pierce Protease Inhibitor EDTA-free Mini Tablet in 10 mL of buffer (Thermo Fisher Scientific A32955, Waltham, MA, USA). The Pierce BCA Protein Assay Kit (Thermo Fisher Scientific 23225) was used to measure protein concentrations. Western blots were performed as reported (O'Donnell et al., 2023), using 70 µg per sample for SCC-13 cells and 40 µg per sample for human tissue and separated on 10% acrylamide SDS-PAGE gels. Gels were transferred to nitrocellulose membranes using an iBlot Gel Transfer Device (Invitrogen) on P3 and blocked for 1 h at room temperature in 1× phosphate-buffered saline (PBS) with 3% bovine serum albumin (BioShop ALB001.100, Burlington, Ontario, Canada). Primary antibody dilutions included anti-PANX1 CT-412 1:1000, anti-PANX3 EL1-84 1:250, anti-PANX3 CT-379 1:1000 (Penuela et al., 2007; Penuela et al., 2009), anti-PANX3 1:100 (Thermo Fisher Scientific 433270) and anti-GAPDH 1:5000 (Millipore Sigma G8795, RRID:AB_1078991, Burlington, MA, USA). Dilutions of 1:10,000 were used for all secondary antibodies, including IRDye-800CW goat anti-rabbit (926-32211, RRID:AB_621843) and IRDye-680RD goat anti-mouse (926-68070, RRID:AB_10956588) IgG secondary antibodies from LI-COR Biosciences (Lincoln, NE, USA). All antibodies were dissolved in 3% bovine serum albumin in 0.05% Tween-20 PBS. PANX1 and PANX3 antibodies were incubated overnight at 4°C, whereas GAPDH and secondary antibodies were incubated for 45 min at room temperature. The LI-COR Odyssey Infrared Imaging System and ImageStudio software (LI-COR Biosciences) were used to image blots and perform protein level and molecular weight quantifications. Protein levels were normalized to GAPDH and presented relative to the control mean value.

## Immunocytochemistry

For analysis of PANX1 localization in SCC-13 cells and SCC-13 cells ectopically expressing a PANX1 plasmid, 300,000 cells were plated on glass coverslips (Thermo Fisher Scientific 1254580, Pittsburgh, PA, USA) in a six-well plate. SCC-13 cells were transfected with 1 μg of PANX1 plasmid 24 h after seeding. For analysis of PANX3-HA localization in SCC-13 cells, 500,000 cells were seeded on glass coverslips in a six-well plate and transfected with 1 μg of PANX3-HA plasmid 24 h after seeding. Forty-eight hours after SCC-13 cells (endogenous PANX1) were plated or 48 h post-transfection for PANX1 or PANX3-HA expressing SCC-13 cells, cells were washed twice with 1× DPBS without $Ca^{2+}$ and $Mg^{2+}$ and fixed with ice-cold methanol–acetone (5:1, v/v) for 15 min at 4°C. For 1 h at room temperature, coverslips were blocked with 10% goat serum (Life Technologies 50062Z) and then incubated overnight at 4°C with anti-PANX1 CT-412 1:500 (Penuela et al., 2007; Penuela et al., 2009) or anti-HA 1:500 (Cell Signaling 3724, RRID:AB_1549585, Danvers, MA, USA) primary antibody in a humidity chamber. After three 5 min washes with 1× DPBS without $Ca^{2+}$ and $Mg^{2+}$, coverslips were incubated with 1:700 Alexa Fluor 488 goat anti-rabbit (Invitrogen A11008, RRID:AB_143165) for 1 h at room temperature. All antibodies were diluted in 1× DPBS without $Ca^{2+}$ and $Mg^{2+}$ with 1% goat serum. After one 1× DPBS without $Ca^{2+}$ and $Mg^{2+}$ and one double-distilled water wash for 5 min each, coverslips were incubated for 5 min at room temperature with Hoechst 33342 (Life Technologies H3570) diluted 1:1000 in double-distilled water to stain cell nuclei. ProLong Gold Antifade Mountant (Thermo Fisher Scientific P36934) was used to mount coverslips. A ZEISS LSM 800 AiryScan Confocal Microscope from the Schulich Imaging Core Facility (University of Western Ontario) was used to image cells at 63× oil magnification with 405 nm (Hoechst 33342) and 488 nm (Alexa Fluor 488) laser lines.

## PANX1 CRISPR/Cas9 deletion

To nucleofect SCC-13 cells (200,000 cells per reaction), the Amaxa P3 Primary Cell 4D-Nucleofector X Kit S (Lonza V4XP-3032, Walkersville, MD, USA) and 4-D Nucleofector X unit (Lonza AAF-1003X) on program DS-138 were used, following the manufacturer's protocol and reagent volumes. To assess nucleofection efficiency cells were co-nucleofected with 0.2 μg pmaxGFP from the P3 kit. Gene Knockout Kit v2-*PANX1* (contains 5′-GAUGGUCACGUGCAUUGGG-3′, 5′-GCCCACGG AGCCCAAGUUCA-3′and 5′-GGCCAGUUGAGGAUG GCCA-3′) and Negative Control sgRNA (mod) 1 (5′-GCACUACCAGAGCUAACUCA-3′) guide RNAs (gRNAs) were purchased from Synthego (Menlo Park, CA, USA), and modified with Synthego's EZ Scaffold to facilitate complexing with Cas9 nuclease. gRNAs were complexed to SpCas9 2NLS Nuclease (Synthego) for 10 min at room temperature and mixed as a 6:1 ratio of gRNA (60 μM) to Cas9 (20 pmol). *PANX1* gRNAs targeted the region immediately following the protein start site in exon one of the *PANX1* gene.

Genomic DNA was isolated from cells using the DNeasy Blood and Tissue Kit (Qiagen 69504, Venlo, the Netherlands), and PCR amplification was performed using Platinum Taq DNA Polymerase High Fidelity (Thermo Fisher Scientific 11304011) both following manufacturer protocols. To assess the success of the CRISPR/Cas9 nucleofection, a PCR was performed using *PANX1* genotyping primers (Table 1), an annealing temperature of 62°C and an extension time of 30 s. PCR products were separated on a 2% agarose gel and imaged using the ChemiDoc XRS+ Gel Imaging System and Quantity One 1-D Analysis Software (Bio-Rad Laboratories, Hercules, CA, USA).

A total of 100,000 nucleofected cells were seeded in 24-well plates immediately after nucleofection to measure confluence, GFP-positive cell counts and cell viability. Twenty-four hours after plating, Incucyte Cytotox Red Dye (Sartorius 4632, Göttingen, Germany) diluted to 250 nM in SCC-13 medium was added to assess cytotoxicity. Images were taken at 10× magnification using the Incucyte S3 Live-Cell Analysis Instrument (Sartorius) at 2 h increments for 4 days using red (400 ms acquisition time), green (300 ms acquisition time) and phase channels. Cell counts per image and percentage confluence were analysed using the Incucyte Cell-By-Cell Analysis Software Module (Sartorius BA-04871).

## PANX1 inhibition

Probenecid (PBN, Invitrogen P36400) and spironolactone (SPL, Selleckchem S4054, Houston, TX, USA) dissolved in water and 100% ethanol (EtOH), respectively, were used to block PANX1 channels. For all cell experiments a final concentration of 1 mM PBN or 20 μM SPL was used, and treatments or corresponding vehicle controls were added to the culture medium on day 0 and replaced every 2 days. A 3-(4,5-dimethylthiazol-2-yl)-2,5-diphenyltetrazolium bromide (MTT) assay (Abcam ab211091, Cambridge, UK) was used to confirm treatments did not affect cell viability. For the MTT assay 50,000 SCC-13 cells were seeded in 12-well plates and treated with a vehicle, PANX1 inhibitor or 1 μM staurosporine (Sigma-Aldrich S6942, Burlington, MA, USA) dissolved in culture media 6 days later. Staurosporine was used as a control for reduced cell viability. Twenty-four hours after treatment MTT was diluted 1:40 in the culture medium and incubated with the cells for 1 h at 37°C and 5% $CO_2$. After the growth

**Table 1. Genotyping and RT-qPCR primers**

| Primer | Forward sequence (5′–3′) | Reverse sequence (5′–3′) | Reference |
|---|---|---|---|
| *PANX1* (genotyping) | AGTCGCTGGGAGCCTGA | CGTAAAATCGCAGCTCACCG | Synthego |
| *Panx3 R1* (genotyping) | CGCAGCATTCCGGAACCTGAG | AACTAGCCGGAGGGGTCG | Abitbol et al. (2019) |
| *Panx3 RT* (genotyping) | CGCAGCATTCCGGAACCTGAG | CTTGTCCTGCGTATGGTG | Abitbol et al. (2019) |
| *PANX1* | AACCGTGCAATTAAGGCTG | GGCTTTCAGATACCTCCCAC | Sayedyahossein et al. (2021) |
| *PANX3* | ATGTGAAAGCTGGAGCCGAA | TTGCCTCACTTGCTCTCTGG | Ye et al. (2012) |
| *β2M* | GTATGCCTGCCGTGTGAACC | AAGCAAGCAAGCAGAATTTGGA | Ye et al. (2012) |

medium was aspirated, 500 μL of dimethyl sulphoxide (DMSO) was added to each well and shaken for 5 min at room temperature to dissolve the formazan products. In a 96-well plate, triplicates of each condition were diluted 1:2 in DMSO, and the OD570 was measured using a VICTOR3 Multilabel Counter plate reader (PerkinElmer, Inc., Shelton, CT, USA). Triplicate values were averaged for each experiment and calculated as a fold change (FC) relative to vehicle averages.

### Growth and migration assays

For growth assays 50,000 SCC-13 cells were seeded in 12-well plates and treated with either vehicle or inhibitor treatment 48 h after seeding. Phase-contrast images were taken at 10× magnification every 6 h for 5 days using the Incucyte S3 Live-Cell Analysis Instrument, and cell counts per image were analysed using the Incucyte Cell-By-Cell Analysis Software Module. For migration assays, 40,000 SCC-13 cells were seeded in Incucyte Imagelock 96-well plates (Sartorius BA-04856) and grown to confluency. Scratch wounds were created in each well using the Incucyte Woundmaker Tool (Sartorius BA-04858), and after one wash with DPBS without $Ca^{2+}$ and $Mg^{2+}$ to remove cell debris, cells were treated with serum free 1× DMEM medium supplemented with either vehicle or inhibitor treatment. Cells were imaged using the Incucyte S3 Live-Cell Analysis Instrument and Incucyte Scratch Wound Analysis Software Module (Sartorius 9600-0012), with phase-contrast images taken every 2 h for 46 h at 10× magnification. Analysis of percentage wound confluence (% wound closure) was performed using the same software.

### DMBA/TPA mouse model

The DMBA/TPA mouse carcinoma protocol (Filler et al., 2007; Halliwill et al., 2016; Sundberg et al., 1997) was employed to investigate the effects of PANX3 on papilloma and carcinoma formation and progression.

Eight-week-old male C57BL/6N wild-type (WT) and congenic global *Panx3* KO mice (Moon et al., 2015) ($N = 5$) were treated with one dose of 25 μg DMBA (dissolved in acetone), followed by twice-weekly treatments of 4 μg TPA (dissolved in acetone) until end-point at 28 weeks. Treatments were administered topically to the first centimetre of the tail at the tail base. Throughout treatment, mice were monitored for their body condition, weight and papilloma formation (volume/growth over time). Papillomas were imaged using an iPhone SE (Apple, Inc., Cupertino, CA, USA) and measured once weekly using a digital calliper according to the University of Western Ontario's CLN-100 Monitoring and Humane Endpoints for Rodent Cancer Models SOP until end-point. Papilloma volumes were calculated using the following equation: $V = 4/3\pi \times (\text{length}/2) \times (\text{width}/2)^2$, where $V$ is the tumour volume, length is the dimension along the longest axis and width is the dimension along the shortest axis. Mice were genotyped as described previously (O'Donnell et al., 2023) using genotyping primers in Table 1, fed Teklad 2018 chow (Envigo, Indianapolis, IN, USA) *ad libitum* and maintained in a 12 h light–dark cycle. At end-point mice were killed using $CO_2$ inhalation (in an increasing concentration).

### Bioinformatics analysis of *PANX1* and *PANX3* expression in cSCC and HNSCC

We utilized the Gene Expression Omnibus (GEO) database to download the RNA-seq FASTQ files of two separate cSCC studies for further validation (Edgar et al., 2002). The GSE191334 database contained eight bulk RNA-Seq samples of cSCC with paired normal controls of human skin organoids, and GSE139505 consisted of the whole transcriptome profile of nine cSCC and seven unmatched healthy skin as controls. Data normalization, dispersion estimation and multiple testing correction were completed using DESeq2 (DOI: 10.18129/B9.bioc.DESeq2) (Love et al., 2014), using

the median-of-ratios method to calculate normalization factors and the Benjamini–Hochberg procedure to control the false discovery rate (FDR) and adjust *P*-values (*P* adj.) to account for multiple testing. The Cancer Genome Atlas (TCGA) biolinks package (Colaprico et al., 2016) was employed to retrieve mRNA expression generated from TCGA HNSCC RNA-sequencing data, which contains 502 HNSCC tissues and 44 adjacent normal tissues, including clinical information and gene expression. A filtering step was applied using TCGAanalyze_Filtering to remove lowly expressed genes and reduce noise, retaining the middle 95% of the data, and incomplete data were deleted before analysis. TCGAanalyze_Normalization was used for normalization based on gene content (gcContent). The exactTest method was used for dispersion estimation, and multiple testing corrections were performed using the FDR with a cut-off of 0.1. Background correction, normalization and expression value calculation were performed on the original datasets using the package Limma of R software (version x64 4.2.1). FC and adjusted *P*-values (*P* adj.) were used to screen differentially expressed genes (DEGs) using DESeq2. $|\log_2(FC)| \geq 1$ and *P* adj.$< 0.05$ were defined as the screening criteria for DEGs. Scatter plots were produced using the Ggplot2 package (https://ggplot2.tidyverse.org/authors.html).

## Patient samples

In collaboration with a local surgical oncologist at the London Regional Cancer Program in London, Ontario, Canada, patient-matched cSCC tumour fragments and normal skin (containing only the epidermis and dermis) samples from 20 different patients as well as 4 additional full-thickness normal skin samples were analysed in this study. The normal skin was either excess tissue from the skin graft used in surgery or obtained via punch biopsy. Patient selection was a prospective, nested cohort of patients presenting from April 2021 to July 2023 with cSCC located on the head and neck. Selection criteria included patients with T2 and T3 solid, non-cystic, non-necrotic tumours of intermediate- to high-risk primary cSCC. Patients were excluded if they presented with extensive nodal metastases or distant disease. cSCC diagnosis was confirmed by a pathologist. Patient demographics are provided in Tables 2 and 3. Images of patient tumours were taken with consent using a password-protected iPhone X (Apple, Inc.).

## Patient demographics

Within our cohort the average age was 75.1 years but ranged from adult (51 years) to geriatric (93 years) patients, with most samples coming from men. Seventy per cent of patients had a history of previous skin cancer, with 30% being immunosuppressed and 35% exhibiting local recurrence. The cSCC tumours were localized to a variety of anatomical locations, with the highest percentages presenting on the scalp (30%), ear

**Table 2. Patient demographics and clinical information**

| CHARACTERISTIC | N = 20 |
| --- | --- |
| Average age (years) | 75.1 |
| Age range (years) | 51–93 |
| Male | 17 (85%) |
| Female | 3 (15%) |
| Immunosuppression | 6 (30%) |
| Previous skin cancer | 14 (70%) |
| Tumour location | |
| Scalp | 6 (30%) |
| Ear | 4 (20%) |
| Cheek | 3 (15%) |
| Temple | 2 (10%) |
| Lip | 2 (10%) |
| Forehead | 1 (5%) |
| Chin | 1 (5%) |
| Neck | 1 (5%) |
| Tumour characteristics | |
| Average size in largest dimension (cm) | 3.2 |
| Largest dimension range (cm) | 0.6–5.7 |
| Local recurrence | 7 (35%) |
| Stage | |
| 1 | 1 (5%) |
| 2 | 1 (5%) |
| 3 | 6 (30%) |
| 4 | 12 (60%) |
| AJCC tumour stage | |
| T1N0M0 | 2 (10%) |
| T2N0M0 | 1 (5%) |
| T3N0M0 | 13 (65%) |
| T3N1M0 | 2 (10%) |
| T3N3aM0 | 1 (5%) |
| T3N0M1 | 1 (5%) |
| Degree of differentiation | |
| Poor | 3 (15%) |
| Moderate | 15 (75%) |
| Well | 1 (5%) |
| *In situ* | 1 (5%) |
| Degree of invasion | |
| Deep | 12 (60%) |
| Perineural | 7 (35%) |
| Lymphovascular | 3 (15%) |
| Normal skin location | |
| Clavicle | 7 (35%) |
| Neck | 5 (25%) |
| Leg | 5 (25%) |
| Arm | 2 (10%) |
| Cheek | 1 (5%) |

Abbreviations: AJCC, American Joint Committee on Cancer; T, tumour; N, node; M, metastasis.

**Table 3. Additional patient demographics, including co-morbidities, medications and clinical outcomes**

| Characteristic | N = 20 |
|---|---|
| **Co-morbidities** | |
| Cardiovascular (HTN, dyslipidaemia, MI, CAD, heart transplant, AAA, arrhythmia, valvular, CHF, angina) | 16 (80%) |
| Endocrine (DM) | 9 (45%) |
| GI (GERD, IBD, CRC, diverticulosis) | 7 (35%) |
| GU (CKD, transplants, bladder cancer, BPH) | 4 (20%) |
| MSK (OA, RA, gout) | 4 (20%) |
| Neurologic (peripheral neuropathy, stroke, hearing loss, Parkinson's disease) | 4 (20%) |
| Haematologic (polycythaemia, anaemia, previous PE, DVT) | 4 (20%) |
| Respiratory (asthma, COPD, ILD) | 3 (15%) |
| Previous cancers (prostate, bladder, colon) | 2 (10%) |
| **Immunosuppression** | |
| Transplant | 2 (10%) |
| Rheumatoid arthritis | 1 (5%) |
| CNS vasculitis | 1 (5%) |
| Crohn's disease | 1 (5%) |
| **Medications** | |
| Metabolic syndrome (anti-hypertensives, statins, anti-platelets, anti-coagulants) | 16 (80%) |
| Diabetes (anti-hyperglycaemics) | 8 (40%) |
| Immune suppressants (steroids, mycophenolate, tacrolimus, methotrexate, biologics) | 6 (30%) |
| Pain control (pregabalin, tramadol, nortriptyline, diclofenac, acetaminophen, oxycocet) | 5 (25%) |
| Anti-microbials (atovaquone, tobramycin) | 1 (5%) |
| **Clinical outcomes** | |
| Adjuvant radiation | 5 (25%) |
| Local recurrence | 5 (25%) |
| Regional recurrence | 4 (20%) |
| Immunotherapy | 3 (15%) |
| Metastasis | 2 (10%) |
| Mortality | 0 (0%) |

Abbreviations: HTN, hypertension; MI, myocardial infarction; CAD, coronary artery disease; AAA, abdominal aortic aneurysm; CHF, congestive heart failure; DM, diabetes mellitus; GI, gastrointestinal; GERD, gastro-oesophageal reflux; IBD, inflammatory bowel disease (Crohn's or ulcerative colitis); CRC, colorectal cancer; GU, genitourinary; CKD, chronic kidney disease; BPH, benign prostatic hypertrophy; MSK, musculoskeletal; OA, osteoarthritis; RA, rheumatoid arthritis; PE, pulmonary embolism; DVT, deep vein thrombosis; DBS, deep brain stimulation; COPD, chronic obstructive pulmonary disease; ILD, interstitial lung disease.

(20%) and cheek (15%) (Fig. 1), and a wide range of tumour sizes were observed. Most tumours were American Joint Committee on Cancer stage T3N0M0 (65%), with a moderate degree of differentiation (75%) and deep invasion (60%). Furthermore, most patients included in this study were classified as having advanced cSCC, due to tumour size and invasion criteria (Migden et al., 2018). Finally, the majority of the normal skin samples were obtained from the clavicle (25%), neck (25%) or leg (25%).

The patients included in this study had a variety of co-morbidities, with cardiovascular and diabetes mellitus being the most common at 80% and 45%, respectively. Consequently, common medications prescribed included those used to treat metabolic syndromes such as anti-hypertensives, statins, anti-platelets, anti-coagulants and anti-hyperglycaemics. The PANX1 channel inhibitor SPL which was originally developed to treat hypertension (Koval et al., 2023) was not prescribed to any patient in the study. Other less common co-morbidities affected the gastrointestinal, genitourinary, musculoskeletal, nervous, haematologic and respiratory systems. In addition to cSCC tumour excision, some patients required adjuvant radiation (25%) and immunotherapy (15%). Patient outcomes included local or regional recurrence (25% for both) and metastasis (10%). Taken together, we believe that our cohort, although small, is representative of the patient population.

## RNA isolation and RT-qPCR

RNA isolation and RT-qPCR for human tissue were performed as described previously (O'Donnell et al., 2023) using a hybrid TRIzol (Life Technologies 15596018) and RNeasy Plus Mini kit (Qiagen 74134) protocol after liquid nitrogen pulverization. A High-Capacity cDNA Reverse Transcription Kit with RNase Inhibitor (Thermo Fisher Scientific 4374966) was used to reverse transcribe 50 ng of RNA as outlined in the manufacturer's protocol in a T100TM Thermal Cycler (Bio-Rad Laboratories). SsoAdvanced Universal SYBR Green Supermix (Bio-Rad Laboratories 1725274) was used to perform RT-qPCR, with samples run in duplicate. Transcript levels were normalized to $\beta$-2 microglobulin ($\beta 2M$), presented as relative to the skin mean value and analysed using the $\Delta\Delta CT$ method in Excel (Microsoft 365, Redmond, WA, USA). See Table 1 for a list of primers used. Logarithmic means were used to perform statistics and report *P*-values, but geometric means were used to construct graphs of figures.

## Immunohistochemistry

cSCC tumour and adjacent skin tissues were fixed in 10% formalin, with routine processing in surgical pathology. Parallel sections (4 µm thickness) were used for haematoxylin and eosin (H&E) and 3,3'-diaminobenzidine (DAB) staining. For DAB staining, tissue slides were incubated at 40°C overnight before xylene deparaffinization (two 5 min and one 3 min incubations) and descending EtOH series rehydration (100% EtOH for 2 min, 100% EtOH for 1 min, 95% EtOH for 2 min, 95% EtOH for 1 min, 70% EtOH for 1 min, distilled water for 2 min). Endogenous peroxidases and alkaline phosphatases were quenched using BLOXALL Endogenous Blocking Solution (Vector Laboratories SP-6000, Newark, CA, USA) for 10 min. Sections were washed with 1× PBS for 5 min, and antigen retrieval was performed with Antigen Unmasking Solution Citrate-based pH 6.0 (Vector Laboratories H-3300) in a Decloaking Chamber (Biocare Medical, LLC, Pacheco, CA, USA) at 112.5°C for 1.5 min, followed by 90°C for 10 s. The slides were cooled by rinsing in tap water and washed with 1× PBS for 5 min. The samples were blocked with 2.5% normal horse serum (Vector Laboratories S-2012-50) for 30 min at room temperature and incubated with 1:250 anti-PANX1 CT-412 diluted in 2.5% normal horse serum at 4°C overnight in a humidity chamber. Secondary antibody control-only samples were left in the blocking solution. The next day slides were washed twice with 1× PBS and incubated with ImmPRESS HRP Horse Anti-Rabbit IgG Polymer Reagent (Vector Laboratories MP-7401) for 30 min at room temperature. After two 5 min washes with 1× PBS, the slides were stained with DAB substrate (Vector Laboratories SK-4100, using reagents 1–3) for 3 min at room temperature. After having rinsed in tap water, the slides were counterstained with Harris haematoxylin (Leica 3801561, Wetzlar, Germany) for 1 min and rinsed again in tap water. Two to three quick dips in 0.6% acid alcohol and 2% ammonium alcohol with rinses in water in between were used to change the colour of Harris haematoxylin to blue. Finally, the slides were cleared with an ascending ethanol series (70% EtOH 1 min, 95% EtOH 1 min, 95% EtOH 2 min, 100% EtOH 1 min, 100% EtOH 1 min, 100% EtOH 2 min) and xylene (two 5 min incubations) and mounted using the Fisher Chemical Permount Mounting Medium (Thermo Fisher Scientific SP15) and microscope cover glass (Thermo Fisher Scientific 12545F). The slides were scanned using a Grundium Ocus40 (Grundium Ltd, Tampere, Finland), and images were processed using QuPath, version 0.5.0 (Bankhead et al., 2017). Objective magnifications of images are outlined in the figure legends. Adobe Photoshop CS6 (Adobe, San Jose, CA, USA) was used to white balance, sharpen and enhance contrast of images, which were annotated by a pathologist.

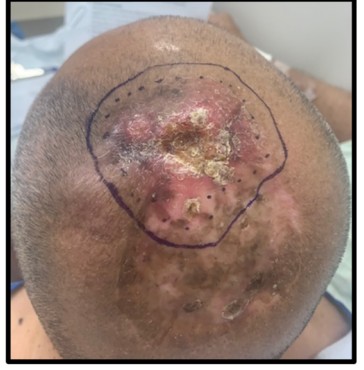 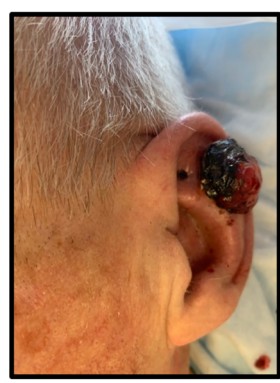 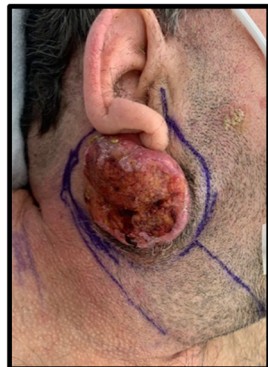

**Figure 1. Images of patient tumours**
Images of patient cSCC (cutaneous squamous cell carcinoma) tumours from the most common anatomical subsites of scalp, ear and cheek (left to right). Images are provided with patient consent.

## Statistics

Statistical analyses were performed using GraphPad Prism 10 (GraphPad Software, San Diego, CA, USA). Outliers were removed from datasets using the outliers test from the same software. Biological and technical replicates and statistical tests used are reported in each figure legend. Based on recommendations from the editorial by Wasserstein et al. to move beyond *P*-values as the only indicator of biological validity to observed differences (Wasserstein et al., 2019), data were referred to in terms of the strength of the evidence when *P*-values did not reach significance. The following *P*-value ranges were used in this study: strong evidence when $P < 0.05$, moderate evidence when $0.05 < P < 0.15$ and weak evidence when $0.15 < P < 0.25$. Any *P*-values $> 0.25$ were considered to show no evidence for differences.

## Results

### PANX1 and PANX3 are present in SCC-13 cells and exhibit distinct localization patterns

To assess PANX1 and PANX3 levels in cSCC, we first analysed PANX1 and PANX3 protein levels in SCC-13 cells. Western blot analysis revealed both PANX1 and PANX3 were detectable in SCC-13 cell lysates (Fig. 2*A*,*B*). Immunocytochemistry using untransfected and PANX1-expressing SCC-13 cells (Fig. 2*C*) revealed that endogenous PANX1 exhibited multiple presentations, where approximately 50% of cell clusters had a predominantly diffuse intracellular localization, whereas the remaining clusters exhibited both cytosolic and prominent plasma membrane localization. In SCC-13 cells exogenously expressing PANX1, the localization pattern was consistent with endogenous PANX1 found in the cytoplasm and plasma membrane, but overexpression seemed to increase the prevalence of intracellular species. Conversely, SCC-13 cells exogenously expressing PANX3-HA exhibited a predominantly punctate intracellular localization when immunolabelled with an anti-HA antibody, although some PANX3-HA could be seen at the plasma membrane (Fig. 2*D*). SCC-13 cells exhibited a low transfection efficiency of approximately 10% for both PANX1 and PANX3-HA constructs. We were unable to evaluate endogenous PANX3 localization in SCC-13 cells due to a lack of human PANX3 antibodies which were suitable for immunocytochemistry.

### Deleting *PANX1* or blocking PANX1 channel function decreases cancer cell properties of SCC-13 cells *in vitro*

We previously demonstrated that reducing PANX1 levels via shRNA knockdown or inhibiting PANX1 channels in another skin cancer reduced the malignant properties of melanoma cells such as growth and migration (Freeman et al., 2019). Because PANX1 was found to be abundant in SCC-13 cells, we evaluated the effects of *PANX1* deletion as well as blocking PANX1 channel function on the corresponding cell properties in this cell line. Here, we ablated *PANX1* in SCC-13 cells using the CRISPR/Cas9 approach, with *PANX1* gRNAs targeting the genomic sequence immediately following the canonical PANX1 start site. PCR-confirmed successful editing occurred in a subset of SCC-13 cells in *PANX1* gRNA mass cultures (Fig. 3*A*), which corresponded to an almost 90% reduction in PANX1 levels ($P < 0.0001$) in *PANX1* gRNA mass cultures compared to Scr controls (Fig. 3*B*,*C*). We found that *PANX1* deletion in *PANX1* gRNA mass cultures significantly reduced SCC-13 growth over time (genotype $P = 0.0091$, time × genotype $P < 0.0001$) compared to Scr mass cultures (Fig. 3*D*,*E*), where the percentage confluence of *PANX1* gRNA mass cultures remained unchanged from 26 to 94 h post-nucleofection. Interestingly, the number of Cytotox Red positive (dying) cells 28 h after nucleofection was unchanged between Scr and *PANX1* gRNA mass cultures ($P = 0.8387$), indicating that the drastically reduced cell numbers observed with *PANX1* ablation resulted from reductions in growth rather than reduced cell viability (Fig. 3*F*). Unfortunately, the growth phenotype with *PANX1* deletion was too severe to perform single-cell colony selection and continue experiments. Thus, we turned to pharmacological inhibition of PANX1 channels instead, which would also be more relevant for translational purposes.

To assess the contribution of PANX1 channel function in SCC-13 cancer cell properties, we utilized previously established PANX1 channel blockers PBN and SPL. Concentrations of 1 mM PBN and 20 μM SPL were selected based on doses previously reported (Freeman et al., 2019; Good et al., 2018) and confirmed to have no detectable effects on cell viability compared to vehicle controls ($P = 0.6045$ and $P = 0.5783$, respectively) via MTT assay (Fig. 4). Consistent with *PANX1* deletion, PANX1 inhibition of SCC-13 cells with both PBN (treatment $P = 0.0045$, time × treatment $P < 0.0001$) and SPL (treatment $P = 0.0066$, time × treatment $P < 0.0001$) treatment significantly reduced cell growth over time compared to water and EtOH vehicle controls. Significant cell growth perturbations were more severe with SPL and evident after only 48 h ($P = 0.0307$) of incubation (Fig. 5*A*,*B*). Next, using scratch assays we investigated SCC-13 migration, determining PANX1 channel inhibition also decreased SCC-13 motility (Fig. 5*C*–*H*). With PBN, although there was no significant treatment effect due to variability ($P = 0.0505$), there was strong evidence to suggest reduced percentage wound closure over time (time × treatment $P < 0.0001$)

compared to water treatment (Fig. 5*C*). This was also evident based on representative images of scratch areas (Fig. 5*E*) and quantifications (Fig. 5*G*) at 24 h ($P = 0.0757$) and 46 h ($P = 0.0066$), indicating blocking PANX1 channels with PBN reduces SCC-13 migration. The same trend with a more pronounced effect was observed with SPL treatment. There was a significant treatment effect with SPL treatment over time (treatment $P = 0.0157$, time × treatment $P < 0.0001$) compared to vehicle control (Fig. 5*D*), where significant reductions in percentage wound closure were evident as early as 16 h ($P = 0.0192$) after initial treatment. Similar to PBN, representative

images (Fig. 5*F*) and quantifications (Fig. 5*H*) at 24 h ($P = 0.0233$) and 46 h ($P = 0.0025$) also indicated reduced SCC-13 migration with SPL-induced PANX1 channel inhibition. Thus, PANX1 promotes the cancerous properties of SCC-13 cells such as growth and migration.

### *Panx3* deletion increases papilloma formation and growth in mouse cutaneous carcinoma model

Due to a lack of PANX3-specific channel blockers and our previous findings that *PANX3* deletion in

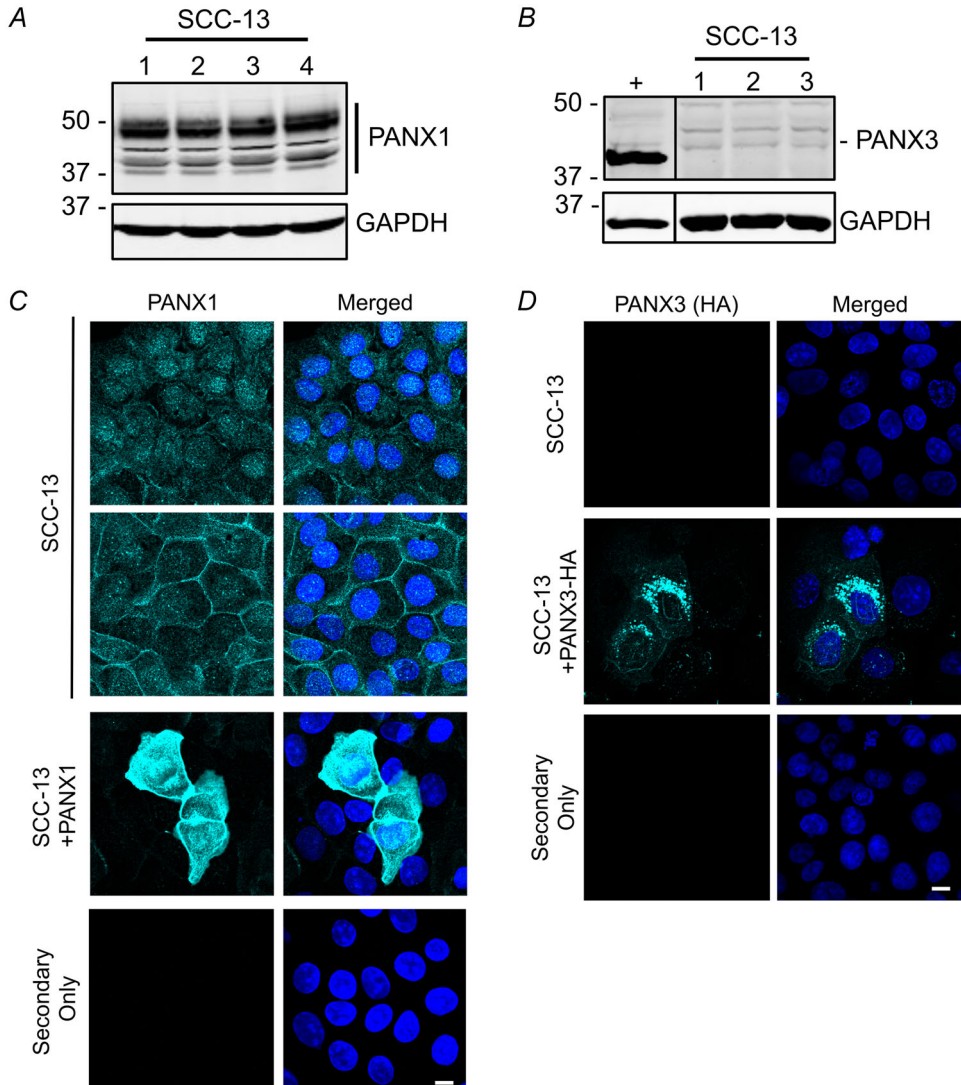

**Figure 2. SCC-13 cells express PANX1 and PANX3 with varying localizations**
Immunoblotting assessed PANX1 (*A*, *N* = 4) and PANX3 (*B*, *N* = 3) protein levels in the SCC-13 cell line. U-2 OS cells ectopically expressing PANX3 (+) as PANX3 positive control. GAPDH as protein loading control, protein sizes in kDa. *C*, Confocal microimages of PANX1 (turquoise) localization in SCC-13 and SCC-13+PANX1 cells (*N* = 3). *D*, Immunofluorescence of PANX3-HA (turquoise, immunostained with anti-HA antibody) localization in SCC-13+PANX3-HA cells (*N* = 3). Untransfected SCC-13 cells as negative control. Nuclei (blue) were counterstained with Hoechst 33342. Coverslips were incubated with Alexa Fluor 488 goat anti-rabbit secondary antibody only as a secondary control. Scale bar: 10 μm. *N* values represent cell line replicates.

isolated primary keratinocytes abrogates cell adhesion (O'Donnell et al., 2023), we utilized the DMBA/TPA mouse cutaneous carcinoma protocol (Filler et al., 2007; Halliwill et al., 2016; Sundberg et al., 1997) on a small subset of male WT and congenic global *Panx3* KO mice ($N = 5$) to investigate the effects of PANX3 on papilloma and carcinoma formation and progression *in vivo*. Eight-week-old mice were treated topically with one dose of 25 µg DMBA, followed by 28 weeks of twice-weekly 4 µg topical TPA treatments at the tail base until end-point at 28 weeks (Fig. 6*A*). Unfortunately, one WT mouse (which developed a papilloma and was included in any analyses of papilloma incidence) died during the treatment period before reaching end-point of causes independent of the study and was therefore removed from all papilloma volume analyses (WT $N = 4$, *Panx3* KO $N = 5$). Most papillomas developed at the site of chemical administration on top of the tail base (Fig. 6*B*, left panel) and were well demarcated, symmetrical and dome shaped without any ulceration, but a few mice

also developed papillomas on the dorsal skin immediately above the tail base, with some papillomas becoming keratinized (Fig. 6*B*, right panel). Of the mice in the pilot study, three of five WT and all five *Panx3* KO mice developed papillomas (Fig. 6*C*), but none developed carcinomas. Generally, *Panx3* KO mice exhibited a papilloma incidence twice that of WT mice, with KO mice developing an average of 2.2 papillomas per mouse compared to the average of 1 papilloma per WT control. Survival curves for time to papilloma onset showed weak evidence for differences between genotypes (Fig. 6*D*, $P = 0.176$); however, the average response latency to first papilloma formation (for mice that developed papillomas) was similar, with WT mice developing papillomas after 25.7 weeks and *Panx3* KO mice developing papillomas after 25.2 weeks. Observation of growth curves for average papilloma volumes in individual mice showed some overlap between genotypes, but most KO mice tended to have increased papilloma growth over time (Fig. 6*E*). Additionally, although no significant genotype effect was

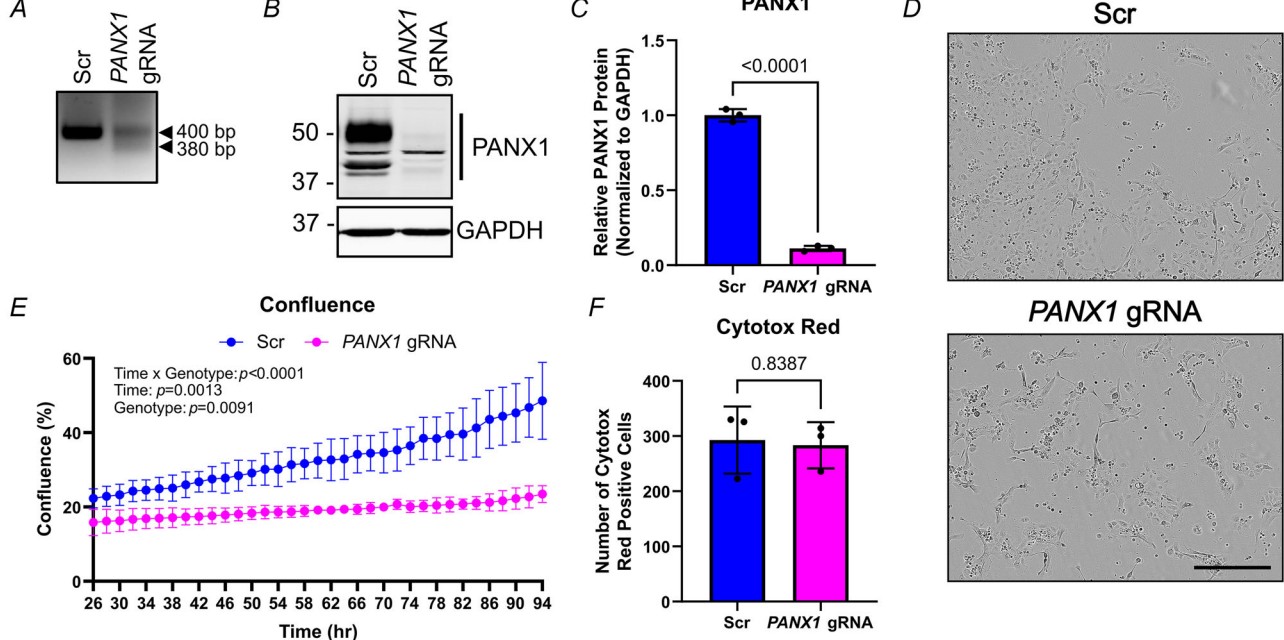

**Figure 3. CRISPR/Cas9 deletion of *PANX1* in SCC-13 cells drastically reduces cell growth**
*A*, PCR genotyping of SCC-13 cell CRISPR/Cas9 mass cultures nucleofected with either *PANX1*-targeted guide RNA (*PANX1* gRNA) or scramble control (Scr) gRNA. Primers produce a 400 bp product in unedited and Scr cells, and a 380 bp product in successfully CRISPR/Cas9 *PANX1* gRNA-edited cells. *B*, Immunoblotting of SCC-13 CRISPR/Cas9 nucleofection mass cultures with Scr or *PANX1* gRNA. GAPDH as protein loading control, protein sizes in kDa. *C*, Quantification of SCC-13 mass cultures shows PANX1 is significantly reduced in *PANX1* gRNA conditions compared to Scr ($P < 0.0001$). Unpaired *t* test. *D*, Phase-contrast images of Scr and *PANX1* gRNA mass cultures taken at 10× magnification 96 h after nucleofection. Scale bar: 400 µm. *E*, Percentage confluence over time for Scr and *PANX1* gRNA SCC-13 mass cultures, beginning 26 h after nucleofection. A significant effect was observed, indicating a reduction in SCC-13 *PANX1* gRNA mass culture growth over time compared to Scr mass cultures (genotype $P = 0.0091$, time × genotype $P < 0.0001$). Two-way repeated measures ANOVA followed by a Sidak's multiple comparisons test. *F*, The number of Cytotox Red positive cells per image in Scr and *PANX1* gRNA SCC-13 mass cultures 28 h after nucleofection is not significantly different ($P = 0.8387$). Unpaired *t* test. Bars indicate mean ± SD. $N = 3$.

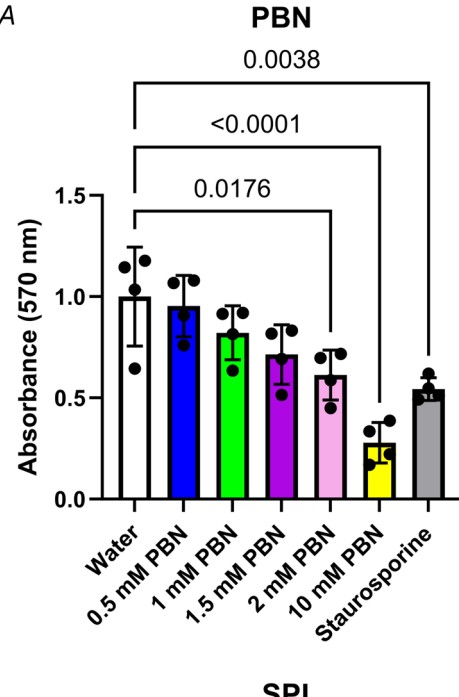

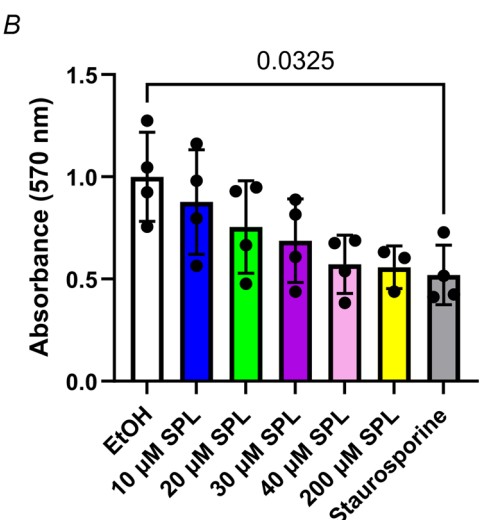

**Figure 4. Analysis of cell viability with PANX1 channel blocker treatment of SCC-13 cells**

An MTT (3-(4,5-dimethylthiazol-2-yl)-2,5-diphenyltetrazolium bromide) assay was used to assess cell viability with increasing concentrations of probenecid (PBN) (*A*) and spironolactone (SPL) (*B*) treatment of SCC-13 cells compared to corresponding water or ethanol (EtOH) vehicle controls. Treatment with 1 mM staurosporine was used as a positive control for reduced cell viability. No significant cytotoxic effects occur at 1 mM PBN ($P = 0.6045$) or 20 µM SPL ($P = 0.5783$) in SCC-13 cells, indicating findings of growth and migration experiments resulted from PANX1 channel blockade, not from changes in cell viability. Two-way ANOVA followed by Sidak's and Tukey's multiple comparisons tests. Bars indicate mean ± SD. $N = 3$.

observed due to high variability between mice (genotype $P = 0.0974$), there was strong evidence to suggest the combined genotype averages for average papilloma volume over time (time × genotype $P = 0.0135$), and the corresponding area under the curve measurements ($P = 0.1106$) was increased in KO mice compared to controls (Fig. 6*F,G*). When analysed at end-point (week 28), the average papilloma volume was found to be significantly increased in *Panx3* KO mice compared to WT mice (Fig. 6*H*, $P = 0.0491$). A power analysis using the earlier results with an α value of 0.05 and a power of 0.8 indicated a sample size of five animals per genotype, indicating our findings were within power. Taken together, this illustrates *Panx3* KO mice tend to have increased papilloma incidence, volumes and growth over time when subjected to a cutaneous carcinogenesis protocol.

## Bioinformatics analysis of *PANX1* and *PANX3* expression in SCC tumours

To explore *PANX1* and *PANX3* transcript expression in patient-derived cSCC tumours compared to normal skin controls, we first performed a bioinformatics search by investigating two cSCC RNA-seq studies available on the GEO database (Fig. 7*A*). In the GSE191334 study, no significant differences were observed in *PANX1* ($P = 0.7165$) or *PANX3* ($P = 0.952$) mRNA expression between tissue types, whereas *PANX1* transcript levels were significantly upregulated ($P = 3.23 \times 10^{-9}$) in cSCC tumours with a FC of +2.65 compared to controls in the GSE139505 study. In the latter study *PANX3* mRNA levels were either not detectable or not reported. Notably, both studies contained small *N* values, and control tissues consisted of either paired human skin organoids or unpaired healthy skin, making it difficult to draw substantial conclusions about *PANX1* and *PANX3* expression in this cohort of patient-derived cSCC tumours. To address the issue of small *N* values, we next evaluated *PANX1* and *PANX3* transcript levels in HNSCC RNA-seq data obtained from TCGA (Fig. 7*B,C*). Although the tissue type of origin differs, with HNSCC arising from squamous mucosal surfaces, both cSCC and HNSCC tumours result from the malignant transformation of epithelial cells and share some key molecular markers (Yan et al., 2011). In this study which contained a much larger *N* value of 502 HNSCC tumours and 44 adjacent normal tissues, *PANX1* mRNA was found to be significantly increased in a subset of HNSCC ($P = 6.54 \times 10^{-14}$), with a FC of +2.20 compared to controls, similar to what was seen in cSCC. *PANX3* transcript levels were not found to be significantly different ($P = 0.5752$) between HNSCC and adjacent tissues but tended to be reduced in HNSCC tumours. Despite the larger *N*

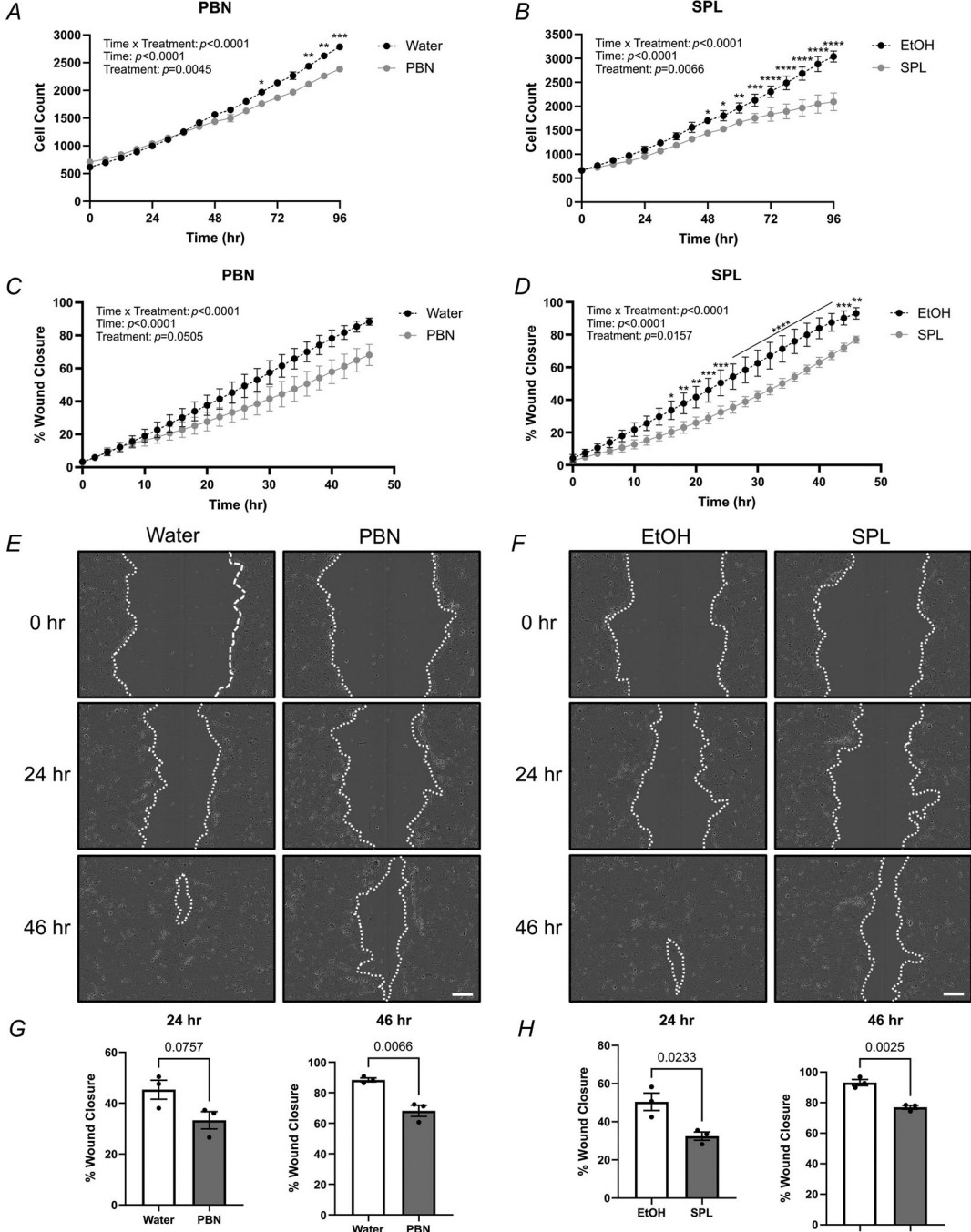

**Figure 5. PANX1 channel inhibition reduces SCC-13 growth and migration**

SCC-13 cell growth was measured over a period of 4 days for cells treated with 1 mM PBN (probenecid) (*A*), 20 µM SPL (spironolactone) (*B*) or the corresponding vehicle control (water and EtOH, respectively). In *A*, *P = 0.0135, **P = 0.0075 at 84 h and **P = 0.0020 at 90 h, ***P = 0.001. In *B*, *P = 0.0307 at 48 h and *P = 0.0180 at 54 h, **P = 0.0059, ***P = 0.0003, ****P < 0.0001. SCC-13 percentage wound closure was measured over a period of 2 days after treatment with 1 mM PBN (*C*), 20 µM SPL (*D*) or the corresponding vehicle control. Two-way repeated-measures ANOVA followed by a Sidak's multiple comparisons test for both growth and migration. In *D*, *P = 0.0192, **P = 0.0054 at 18 h, **P = 0.0020 at 20 h and **P = 0.0013 at 46 h, ***P = 0.0006 at 22 h, ***P = 0.0002 at 24 h and 44 h, ****P < 0.0001. Representative phase-contrast images of scratch wounds in SCC-13 cells treated with 1 mM PBN (*E*), 20 µM SPL (*F*) or the corresponding vehicle control at 0, 24 and 46 h timepoints. Dotted white line outlines scratch. Scale bar: 200 µm. *G*, Quantifications of 24 and 46 h timepoints for SCC-13 cells treated with either water or 1 mM PBN exhibited a reduced trend at 24 h (*P = 0.0757*) and significant

reduction for percentage wound closure at 46 h ($P = 0.0066$) with PBN treatment. *H*, SPL (20 µM) treatment of SCC-13 cells significantly reduced percentage wound closure at 24 h ($P = 0.0233$) and 46 h ($P = 0.0025$). Unpaired *t* test. Bars indicate mean ± SD. $N = 3$.

values of this study, *PANX1* and *PANX3* mRNA were detectable only in 31 normal tissue and 89 HNSCC tumour samples.

### In human cSCC tumours *PANX3* mRNA is reduced compared to matched skin controls, whereas PANX1 is upregulated and present within all regions of the tumour microenvironment

We wanted to investigate whether the PANX1 and PANX3 expression patterns of our patient-derived cSCC and normal skin cohort were consistent with those seen previously (Cowan et al., 2012; Halliwill et al., 2016). Beginning with PANX3 we determined that in human full-thickness skin samples, PANX3 was highly abundant (Fig. 8*A*), and *PANX3* transcripts were significantly reduced ($P = 0.0452$) in cSCC tumour fragments compared to patient-matched skin samples containing the epidermal and dermal layers (Fig. 8*B*). Overall, this indicates that *PANX3* transcripts are decreased in human cSCC tumours compared to patient-matched unaffected epidermis and dermis.

cSCC tumour fragments are three-dimensional structures that contain other cell types in addition to cancer cells such as cancer-associated fibroblasts, immune infiltrate and blood vessels. Previous work from our group has also showed that in melanoma, PANX1 localizes to multiple regions within the tumour structure (Freeman et al., 2019). Thus, to localize PANX1 expression within the various cells of the cSCC tumour microenvironment, we performed immunohistochemistry of tumours and adjacent skin tissue from the most common anatomical subtypes of the scalp, ear and cheek (Figs 9 and 10). According to pathologist observations PANX1 is present throughout all regions of cSCC tumours and adjacent skin (Fig. 9*B*), with relative staining intensity slightly increased in the tumour. Within the tumour PANX1 staining was present in both tumour nests and stromal regions but seemed to be more prominent within the cSCC cancer cells as opposed to the tumour-infiltrating lymphocytes and stromal fibroblasts present in the stroma surrounding tumour nests (Fig. 9*C,D*). However, PANX1 was visibly reduced in keratinized regions and necrotic tissue as expected (Fig. 10*A,B*). In the adjacent skin, PANX1 was present throughout all the living layers of the epidermis, with relatively higher levels in basal keratinocytes (Fig. 9*E*). Visually increased PANX1 levels were also observed within blood vessels (Fig. 9*D*) and nerve bundles (Fig. 10*C*). Overall, multiple cell types in

cSCC tumours and adjacent skin are positive for PANX1 protein expression.

To obtain a more quantitative understanding of the relative PANX1 levels in cSCC and normal skin controls, we analysed our cohort of cSCC and non-adjacent skin samples for PANX1 transcript and protein expression. We used non-adjacent normal skin samples as our control for this analysis, because it has been previously shown that PANX1 levels transiently upregulate at the site of cutaneous wounds in mice (Penuela et al., 2014), and thus tumour-adjacent skin may have increased PANX1 levels in response to the cSCC tumour. In human full-thickness skin samples, PANX1 was present at low levels (Fig. 11*A*). Despite a lack of differences seen in *PANX1* mRNA levels (Fig. 11*B*, $P = 0.9254$) between tissue types, immuno-blotting exhibited a significant increase in cSCC tumour PANX1 levels compared to patient-matched normal skin (containing epidermis and dermis only) controls (Fig. 11*C,D*, $P = 0.0011$). Our cohort of patient-matched samples (Fig. 11*E*) revealed most sample pairs exhibited the pattern of higher PANX1 in cSCC tumours, but interpatient variability was noticeable for PANX1 levels in both skin and cSCC samples. We also assessed how PANX1 levels may correlate with tumour recurrence and size. Because 70% of patients had a previous incidence of skin cancer, we analysed PANX1 expression in tumours which developed at the site of previous skin cancer (local recurrence) or were present at a novel anatomical location (distant recurrence) but found no significant differences in PANX1 levels between sites (Fig. 11*F*, $P = 0.3323$). Patient tumours were also stratified based on tumour volume and the largest tumour dimension, dividing the 16 cSCC samples in half based on the median value (11.24 cm$^2$ and 3.7 cm, respectively) to create below-median and above-median groups. Although no differences were evident based on tumour volume (Fig. 11*G*, $P = 0.8734$), cSCC patient tumours with the largest dimension above the median value had significantly higher PANX1 levels compared to tumours with their largest dimension below the median (Fig. 11*H*, $P = 0.0377$). Overall, this indicates PANX1 is increased in human cSCC tumours, where it localizes to all regions within the cSCC tumour microenvironment and is associated with a larger tumour size.

### Discussion

In this study we found that PANX1 and PANX3 show opposite expression pattern changes in cSCC, with PANX1 increased and PANX3 decreased in this type

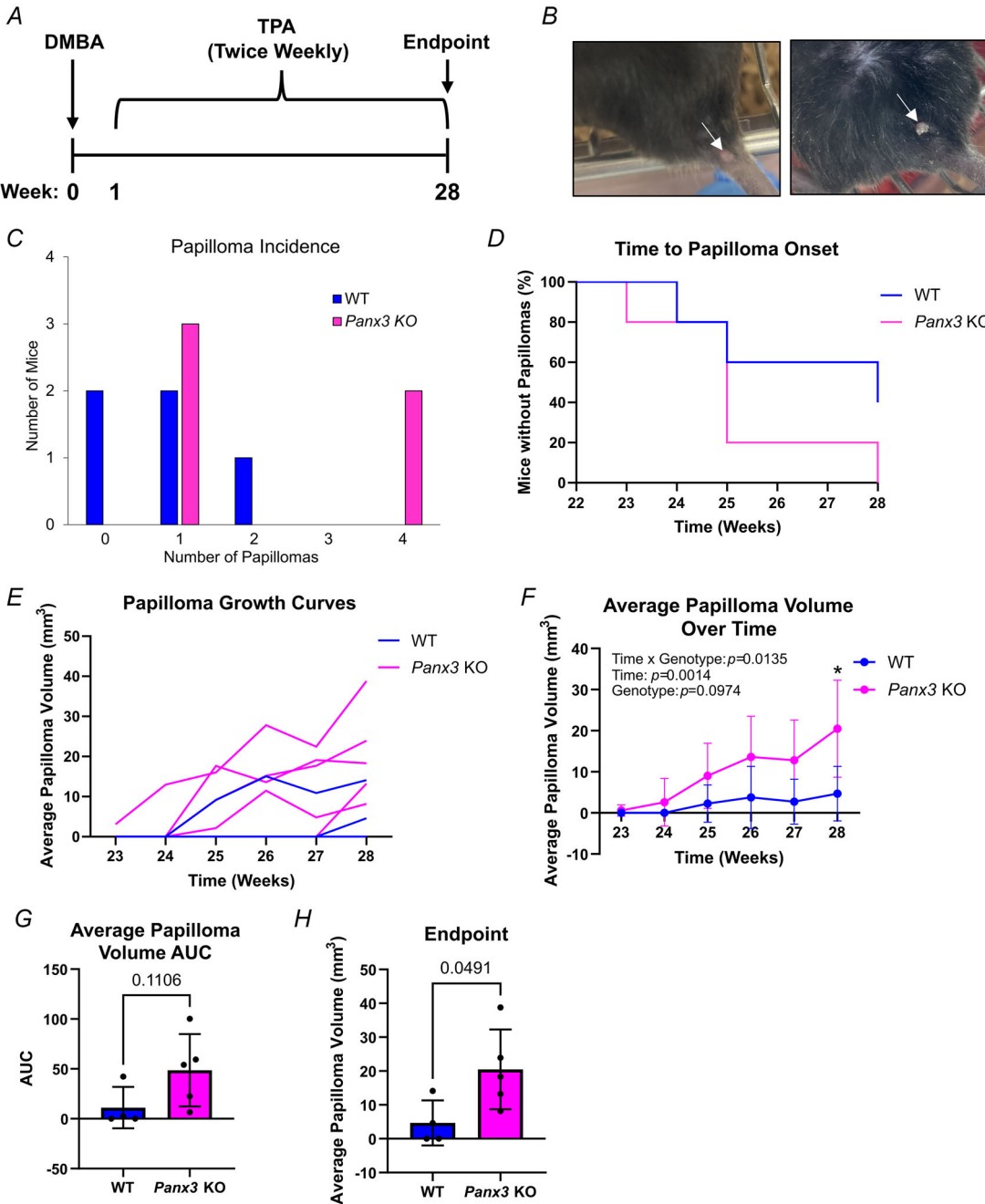

**Figure 6. DMBA/TPA (7,12-dimethylbenz(a)anthracene/12-otetradecanoylphorbol-13-acetate)–treated *Panx3* KO (knockout) mice tend to have increased papilloma incidence, volumes and growth over time**
*A*, Timeline of experimental design. Eight-week-old WT (wild-type) and *Panx3* KO (knockout) mice were subjected to one dose of 25 μg DMBA treatment (week 0), followed by 28 weeks of twice-weekly treatments of 4 μg TPA. Twenty-eight weeks after the initial DMBA treatment, mice were killed. *B*, Images of papillomas that developed on treated mice. Papilloma indicated by white arrow. *C*, KO mice had a greater average papilloma incidence of 2.2 papillomas per mouse compared to the average of 1 papilloma per WT control, with three of five WT and all five KO mice developing papillomas. *D*, Survival curves for time to papilloma onset were similar between genotypes ($P = 0.176$). Log-rank test. $N = 5$ for WT, KO mice for panels *A–D*. *E*, Papilloma growth curves for average papilloma volume over time. Individual lines show measurements per mouse. *F*, Although only significant at week 28 (*$P = 0.0416$), average papilloma volume showed moderate evidence (genotype $P = 0.0974$) to an increase in KO mice, which was significant over time (time × genotype $P = 0.0135$). Two-way repeated-measures ANOVA and multiple unpaired *t* tests. *G*, Area under the curve (AUC) measurements for average papilloma volume over time showed moderate evidence to an increase in KO mice that was not statistically significant ($P = 0.1106$). Unpaired *t* test. *H*, Average papilloma volume at end-point (week 28) was significantly increased in KO mice compared to controls ($P = 0.0491$). Unpaired *t* test. $N = 4$ for WT, $N = 5$ for KO in panels *E–H*. Bars represent mean ± SD.

of keratinocytic tumour compared to patient-matched healthy non-adjacent skin controls. We also determined that in full-thickness normal skin, both PANXs are expressed, with PANX1 being detected at low levels and PANX3 highly abundant, consistent with reports of low PANX1 and high PANX3 expression in the dorsal skin of aged mice (O'Donnell et al., 2023; Penuela et al., 2014). In each tissue type, both PANXs exhibit opposite expression patterns, most likely due to their distinct homeostatic actions in normal skin. Previous findings from our group suggest PANX1 plays more of a proliferative and developmental role in the skin, particularly in keratinocytes of which cSCC develops. PANX1 is prevalent in young murine skin, and basal keratinocytes where it traffics to the cell surface, but normally exhibits an age-related decline and reduction during keratinocyte differentiation where it changes its localization pattern

to become more cytosolic (Penuela et al., 2014). This notion is consistent with the cSCC cancer condition where cells are aberrantly proliferative and their ability to terminally differentiate is compromised. Conversely, PANX3 functions in maintaining epidermal structure and integrity and is present at high levels in aged skin and differentiated keratinocytes where it was found to have an intracellular localization (Cowan et al., 2012; O'Donnell et al., 2023; Zhang et al., 2019; Zhang et al., 2021). This indicates the protein is critical in homeostatic ageing processes to ensure proper keratinocyte turnover within the epidermis and therefore must be downregulated for cSCC carcinogenesis. Additionally, ablating *Panx3* in mice was found to increase inflammatory signalling in the neonatal epidermis and the incidence of dermatitis in an aged cohort (O'Donnell et al., 2023). Because an inflammatory microenvironment has been shown to

**A** cSCC

| Study | Transcript | Log2 Fold Change | Fold Change | *p* value | *p* adj | *N* value Tumour | *N* value Control |
|---|---|---|---|---|---|---|---|
| GSE191334 | *PANX1* | 0.3187 | +1.25 | 0.2190 | 0.7165 | 8 | 8 |
| GSE191334 | *PANX3* | 1.0940 | +2.13 | 0.7188 | 0.9952 | 8 | 8 |
| GSE139505 | *PANX1* | 1.4062 | +2.65 | $2.35 \times 10^{-10}$ | $3.23 \times 10^{-9}$ | 9 | 7 |
| GSE139505 | *PANX3* | - | - | - | - | 9 | 7 |

**B** HNSCC

| Transcript | Log2 Fold Change | Fold Change | FDR | *p* value | *N* value Tumour | *N* value Control |
|---|---|---|---|---|---|---|
| *PANX1* | 1.1368 | +2.20 | $1.54 \times 10^{-12}$ | $6.54 \times 10^{-14}$ | 502 | 44 |
| *PANX3* | -0.3453 | -1.27 | 0.6549 | 0.5752 | 502 | 44 |

**C** HNSCC

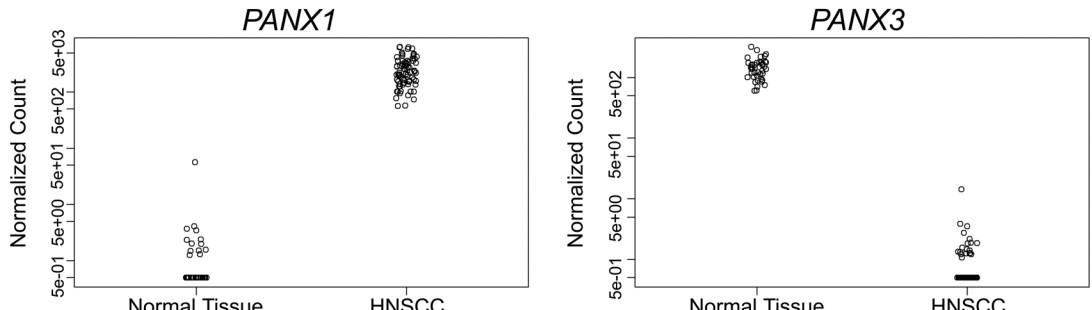

**Figure 7. *PANX1* mRNA is upregulated in cSCC (cutaneous squamous cell carcinoma) and HNSCC (head and neck SCC) tumours compared to corresponding control tissue**

*A*, Table shows the normalized expression levels of *PANX1* and *PANX3* mRNA in cSCC samples compared to normal skin control samples (either human skin organoids or unmatched healthy controls). RNA-seq expression data were derived from the GEO (Gene Expression Omnibus) gene expression studies GSE191334 and GSE139505. *P* adj., *P*-value adjusted. *B*, Table outlines the normalized expression levels of *PANX1* and *PANX3* in HNSCC tumour samples compared to normal adjacent tissue controls. Data were obtained from TCGA (The Cancer Genome Atlas) database. FDR, false discovery rate. *C*, The *z*-score normalized transcript expression levels of *PANX1* and *PANX3* in TCGA_HNSCC RNA-seq samples compared to normal tissue show a marked increase in *PANX1* ($P = 6.54 \times 10^{-14}$) but a decrease in *PANX3* ($P = 0.5752$) in HNSCC tumours compared to controls. Out of 44 control and 502 tumour samples, *PANX1* and *PANX3* transcript levels in the HNSCC and normal tissue were detectable in 31 normal tissue and 89 HNSCC tumour samples.

facilitate cutaneous carcinogenesis (Neagu et al., 2019), reduced PANX3 levels in the skin could promote a pro-tumorigenic environment.

The PANX1 expression findings in our cSCC patient cohort conflict with earlier findings from our group (Cowan et al., 2012) but are consistent with the bioinformatics data from the Human Protein Atlas, and cSCC and HNSCC RNA-seq study data mined from GEO and TCGA, respectively. Inconsistencies were likely due to a smaller sample size in the previous study but may also be due to differences in the tumour stage, where most of the tumours included in this study were advanced. Because PANX1 is upregulated in most other cancer types in which it has been reported, including melanoma, breast cancer and leukemia (Laird & Penuela, 2021), it seems more likely that PANX1 is increased in cSCC compared to normal tissue. For PANX3, trends seen in cSCC corresponded to those observed in DMBA/TPA-induced murine papillomas and carcinomas (Halliwill et al., 2016), previous immunofluorescence microscopy findings from

our group (Cowan et al., 2012) and HNSCC *in silico* findings, where *Panx3*/PANX3 levels were reduced in each case compared to controls.

We determined that PANX1 channel activity promoted cancer cell properties such as growth and migration in SCC-13 cells using established PANX1 channel inhibitors PBN and SPL. PANX1 channel activity in SCC-13 cells resembles that of human melanoma cells, whereby blocking PANX1 channels also reduced melanoma cell numbers and motility (Freeman et al., 2019). Although both channel blockers exhibited the same trends, reductions in SCC-13 tumorigenic properties were more prominent with SPL treatment, most likely due to the increased specificity and potency and decreased off-target effects when inhibiting PANX1 channels with SPL rather than PBN (Koval et al., 2023). However, *PANX1* deletion in SCC-13 mass cultures decreased cell growth to a greater extent than channel blockade. This suggests PANX1 action through its signalling and inter-actome also contributes to cancer cell properties, similar to the interplay between PANX1 and the canonical Wnt pathway in melanoma (Sayedyahossein et al., 2021). Our finding that PANX1 levels are increased in larger cSCC tumours also corresponds with our SCC-13 PANX1 deletion and inhibition results and suggests that PANX1 channel-dependent and PANX1 channel-independent functions may influence cSCC cell proliferation, indicating a role for PANX1 in cSCC tumour progression. PANX1 was found to localize to both the cell surface and intracellularly in SCC-13 cells, indicating it could act as both an ATP release channel at the plasma membrane and a calcium leak channel in the endoplasmic reticulum – both of which would be sensitive to PBN and SPL inhibition – to promote tumorigenic properties like growth and migration. PANX1 was prevalent in many tumour cell types in addition to the cSCC cancer cells such as in blood vessels and immune cells, meaning cross-talk between the different cell types within the tumour microenvironment may be possible. Future studies could investigate the potential role of PANX1 in immune infiltration/evasion and angiogenesis in cSCC tumours. As for PANX3, we determined that exogenously expressed PANX3-HA protein localized predominantly intra-cellularly in SCC-13 cells, with a small pool trafficking to the cell surface, but we are unsure of the specific organelle compartments where PANX3 is present. Zhang et al. (2019) previously showed that in HaCaT keratinocytes, PANX3 overexpression increased extracellular ATP release and $Ca^{2+}$ release from the endoplasic reticulum, so it is possible these homeostatic functions in keratinocytes are lost with reductions in PANX3 to promote cSCC formation. However, because our assessment of the effects of PANX3 on cSCC formation was performed using a global *Panx3* KO mouse model, it remains to be seen whether the reduced papilloma incidence and

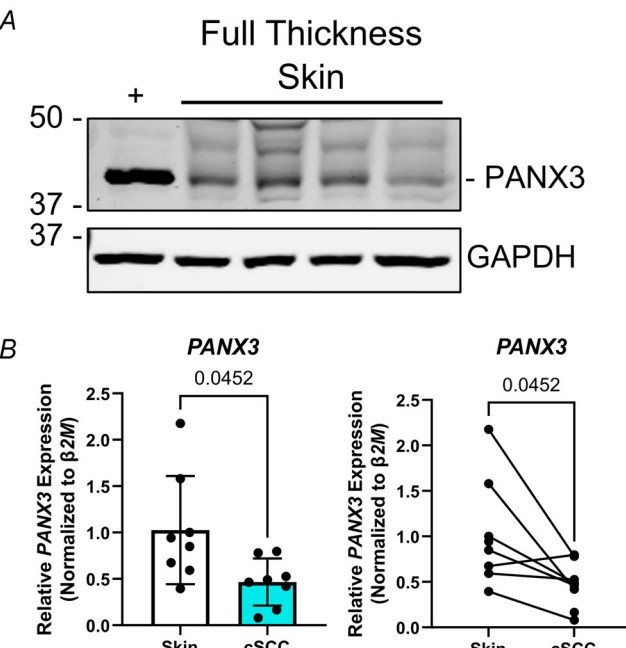

**Figure 8. *PANX3* mRNA levels are reduced in cSCC (cutaneous squamous cell carcinoma) tumour fragments compared to patient-matched skin**

*A*, Immunoblotting indicates PANX3 is present in full-thickness human skin samples (*N* = 4). U-2 OS cells expressing a PANX3 plasmid as positive control for PANX3 (+). GAPDH used as protein loading control, protein sizes in kDa. *B*, *PANX3* transcripts were significantly reduced (*P* = 0.0452, *N* = 8) in cSCC tumour fragments compared to normal skin (minus hypodermis) via RT-qPCR. β-2-Microglobulin (*β2M*) was used to calculate normalized mRNA using the ΔΔCT method. Paired *t* test. Bars represent mean ± SD. Scatter plots with bars show mean values, and before–after plots show patient sample matching.

growth with *Panx3* ablation were due to reductions in PANX3 activity, signalling and/or protein interactions. Taken together, this indicates that in cSCC, PANX1 may exhibit pro-tumour activities and PANX3 may exhibit anti-tumour activities.

Despite cSCC being one of the most commonly diagnosed skin cancers with yearly increases in incidence estimated at 3%–8% worldwide since the 1960s (Abbas & Kalia, 2015), there are very few tools to study this

disease. In our study, experiments were performed using one of the only available cSCC cell lines (SCC-13), which we understand as a limitation because this singular cell line may not be representative of the patient population. Moreover, SCC-13 cells were derived from a carcinoma *in situ* (Rheinwald & Beckett, 1981), which is an early, more benign form of cSCC (Adalsteinsson et al., 2021) that more closely resembles a precancerous papilloma lesion (as seen in the mouse carcinoma model) than an advanced

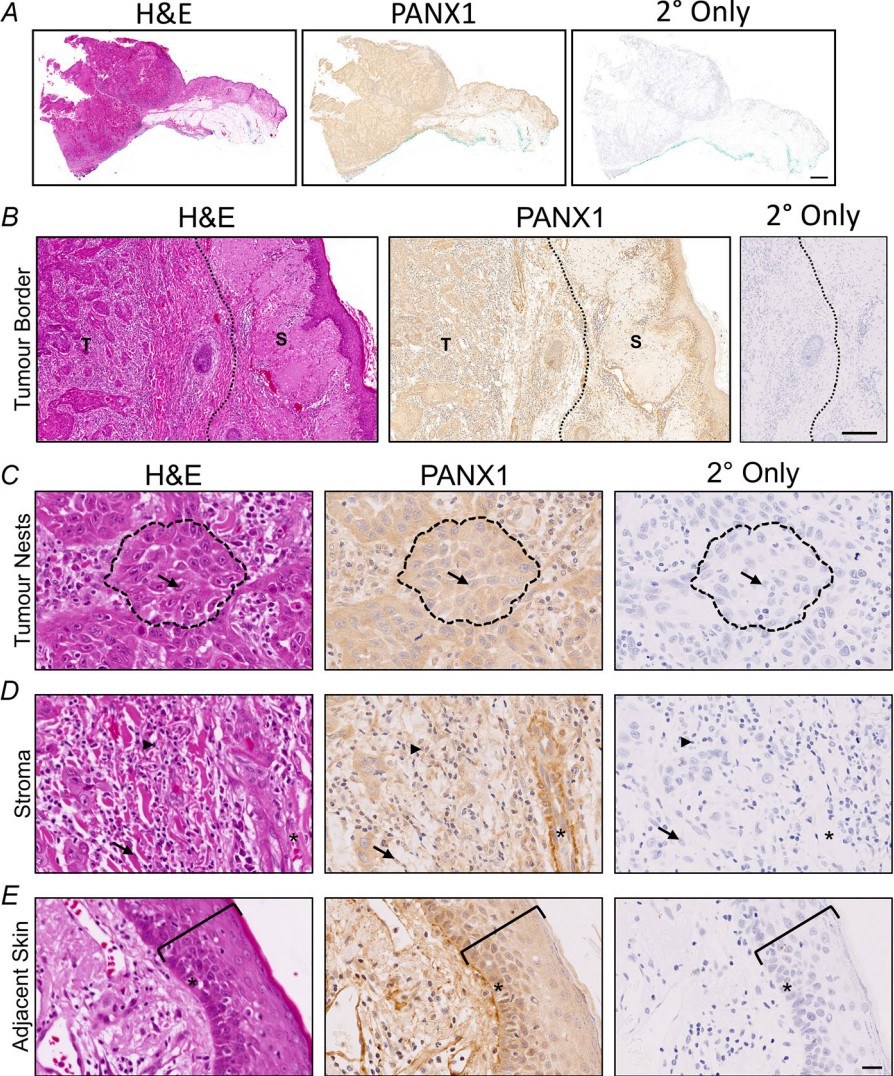

**Figure 9. PANX1 localizes to all cSCC tumour regions**
Representative images for haematoxylin and eosin (H&E)–stained patient-derived cSCC tumour fragment and adjacent skin tissue, with matching immunohistochemistry [brown DAB (3,3′-diaminobenzidine) stain] indicating PANX1 localization and secondary-only control (2° Only). Nuclei (blue) counterstained with Harris haematoxylin. *N* = 8. *A*, Whole sample view of cSCC tumour and adjacent skin. Objective magnification: 0.35×. Scale bar: 2 mm. *B*, Border of cSCC tumour (T) and adjacent skin (S), with division between tissue types denoted by a dotted line. Objective magnification: 6.26×. Scale bar: 200 μm. *C*, Carcinoma cells contained within tumour nests (outlined by dotted line) express PANX1. The nucleus of a carcinoma cell is denoted by an arrow. *D*, The stromal region of the cSCC tumour contains tumour-infiltrating lymphocytes (indicated by arrowhead), cancer-associated fibroblasts (indicated by arrow) and blood vessels (lumen denoted by an asterisk), which also express PANX1. *E*, In the tumour-adjacent skin, PANX1 is present throughout all layers of the epidermis. Epidermal layers are bracketed, with asterisk (*) indicating the basal layer of keratinocytes. Objective magnification: 40×, scale bar: 50 μm for *C–E*.

cSCC (our cSCC patient sample tumours). Furthermore, there is a lack of epidemiological and expression data for NMSCs, as evidenced by the insufficient bioinformatics data available to be mined for this study, which is most likely due to the low fatality with NMSCs and lack of a NMSC surveillance system or any sort of record keeping of cSCC incidence in many countries (Abbas & Kalia, 2015). This reality has further impeded research into understanding this disease and was part of the rationale for performing this study.

For the chemical carcinoma model, we used our previously established *Panx3* KO mouse model (Moon et al., 2015) to assess the potential role of PANX3 in cSCC carcinogenesis because currently there are no commercially available PANX3-specific channel blockers. Unfortunately, our mice were bred with a C57BL/6 background that has previously shown to be more tumour resistant to the DMBA/TPA protocol (Sundberg et al., 1997). This was evident in our study because none of the treated mice developed carcinomas, which we recognize

as a limitation. We were able to include only $N = 5$ mice per genotype but opted to use male mice (instead of females which are normally suggested) (Filler et al., 2007) to be more representative of the human patient population because men account for a much higher percentage of cSCC incidence than women (Urban et al., 2021). However, because our results were within power, we did not have reason to increase the number of mice used in this study. Finally, we opted to administer DMBA/TPA treatments to the tail skin of mice to avoid shaving the dorsal skin with every treatment, which would irritate or damage the epidermis and potentially affect interpretation of results. Furthermore, murine tail skin is more similar to human thin skin because it contains less hair follicles and has a thicker epidermis than that of mouse dorsal skin, making it a more suitable selection to model human cSCC carcinogenesis (Xie & Chen, 2022).

Overall, through an *in vitro* investigation of PANX1 deletion and channel blocking, an *in vivo* characterization of a chemical carcinoma model in *Panx3* KO mice and

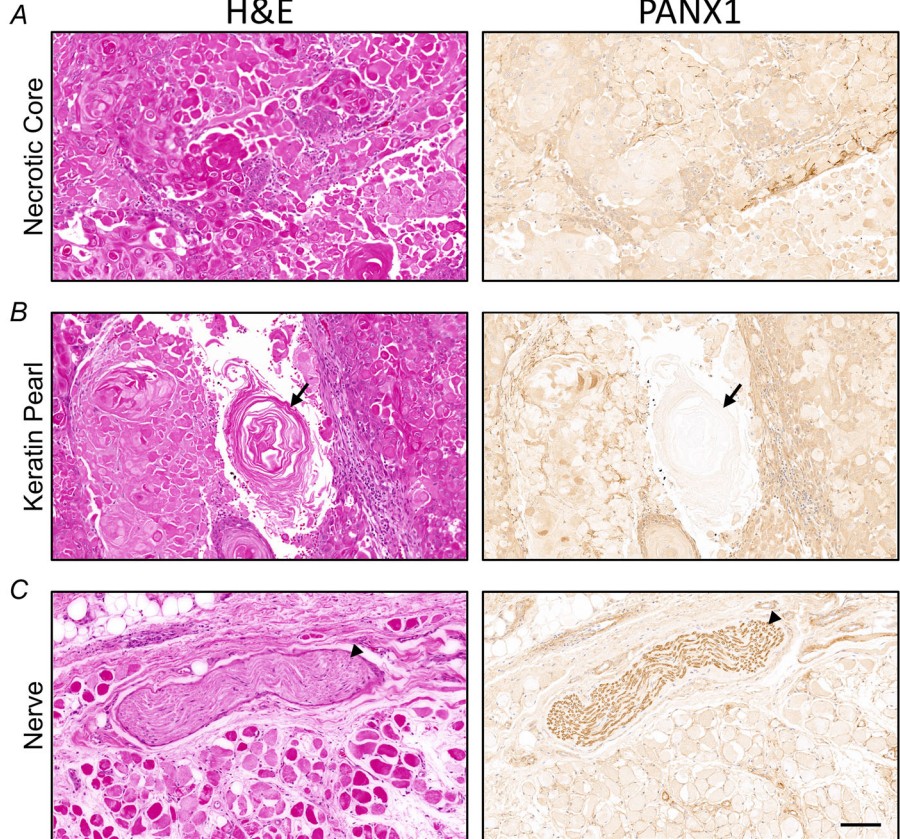

**Figure 10. Positive and negative controls for PANX1 immunohistochemistry of cSCC (cutaneous squamous cell carcinoma) tumours and adjacent skin**
Representative haematoxylin and eosin (H&E) and PANX1 immunohistochemistry [DAB (3,3′-diaminobenzidine)] stain in brown] images of structures within cSCC tumour and adjacent skin. *A*, Necrotic core contains mostly keratin and dying or dead cells. *B*, Arrow indicates keratin pearl within cSCC tumour, which does not stain for PANX1. *C*, Arrowhead indicates nerve in adjacent skin, which stains strongly for PANX1. Objective magnification: 11.83×. Scale bar: 100 μm. Nuclei shown in blue.

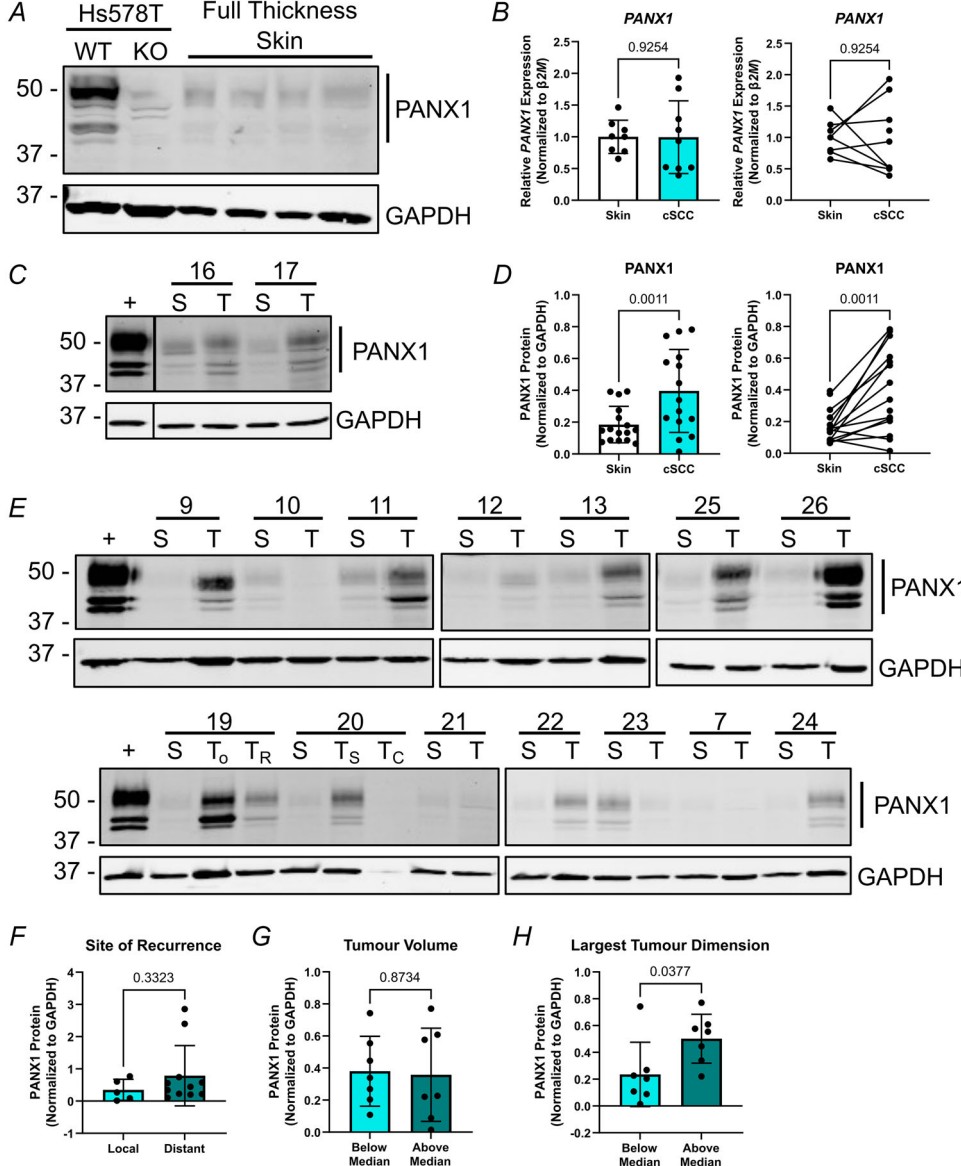

**Figure 11. PANX1 protein levels are upregulated in human cSCC (cutaneous squamous cell carcinoma) tumour fragments compared to normal skin**

*A*, Western blotting showed PANX1 is present at very low levels in full-thickness normal human skin ($N = 4$). Hs578T WT (wild-type) cells as positive control and Hs578T *PANX1* KO (knockout) cells as negative control. *B*, RT-qPCR revealed no differences in *PANX1* transcripts between normal skin (minus hypodermis) and patient-matched cSCC tumour fragments ($P = 0.9254$, $N = 9$). $\beta$-2-Microglobulin ($\beta 2M$) was used to calculate normalized mRNA using the $\Delta\Delta$CT method. *C*, Immunoblotting assessed PANX1 levels in patient-matched normal skin (S, minus hypodermis) and cSCC tumour fragments (T) where each # corresponds to a different patient. *D*, Quantification showed PANX1 levels are significantly increased in cSCC tumour fragments compared to normal skin ($N = 16$, $P = 0.0011$). *E*, Western blots of all samples used for PANX1 quantification (*C*) show variability between patients. Quantifications excluded degraded tumour samples (and corresponding normal skin), evidenced by a lack of GAPDH expression (protein loading control). GBM17 cells as positive control for PANX1 (+). $T_o$, original tumour; $T_R$, recurrent tumour; $T_S$, tumour from scalp; $T_C$, tumour from cheek. Protein sizes in kDa. *F*, PANX1 levels were not significantly different ($P = 0.3323$, $N = 16$) in cSCC tumour fragments developing at local or distant recurrence sites from patients with a previous incidence of skin cancer. *G*, cSCC tumours above and below the median tumour volume of 11.24 cm$^2$ showed no differences in PANX1 levels ($P = 0.8734$, $N = 14$). *H*, cSCC tumours above the median largest tumour dimension of 3.7 cm had significantly increased PANX1 compared to those below the median value ($P = 0.0377$, $N = 14$). Paired *t* test. Bars represent mean $\pm$ SD. Scatter plots with bars show mean values, and before–after plots show patient sample matching.

an analysis of patient-derived tissue, we observed that PANX1 and PANX3 dysregulation may have potential tumour-promoting or tumour-suppressive effects for keratinocyte transformation into cSCC, respectively. These findings offer the potential for PANX1 and PANX3 as new therapeutic targets in cSCC treatment, specifically for advanced disease.

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

## Additional information

### Data availability statement

All raw data are available upon request.

## Competing interests

The authors have no competing interests to declare.

## Author contributions

Conceptualization: B.L.O'D., D.J., L.D., S.K.L., K.R., S.P.; data curation: Z.K.; formal analysis: B.L.O'D., Z.K.; funding acquisition: S.P.; investigation: B.L.O'D., D.J., A.Bh., Z.K., D.S., A.By., S.S.; methodology: B.L.O'D., D.J., S.P.; project administration: B.L.O'D., D.J., S.P.; resources: S.K.L., M.C., K.R., S.P.; software: Z.K.; supervision: L.D., S.K.L., K.R., S.P.; visualization: B.L.O'D., A.Bh., Z.K., D.S., A.By., M.C.; writing – original draft: B.L.O'D., S.P.; writing – review and editing: all authors.

## Funding

This work was supported by a Canadian Institutes of Health Research Project grant (FRN 153112) and a London Regional Cancer Program Catalyst Grant to S.P.

## Acknowledgements

The authors thank Dr Rafael Sanchez-Pupo and Dr Brent Wakefield (University of Western Ontario) for their assistance with statistical analyses, and Dr John Kelly (University of Western Ontario) for his assistance with the nucleofections. They would also like to thank the Pathology and Laboratory Medicine Translational Research Service at London Health Sciences Centre for providing tumour tissue slides and Linda Jackson-Boeters (University of Western Ontario) for her assistance with the DAB staining protocol. Finally, the authors thank Dr Kathryn Roth's patients who provided samples for this study.

## Keywords

cutaneous squamous cell carcinoma, PANX1, PANX3, skin

## Supporting information

Additional supporting information can be found online in the Supporting Information section at the end of the HTML view of the article. Supporting information files available:

**Peer Review History**

