## [Peer Review History · The Journal of Physiology]

Pannexin 1 and Pannexin 3 differentially regulate the cancer cell properties of cutaneous squamous cell carcinoma

Brooke L O'Donnell, Danielle Johnston, Ayushi Bhatt, Zahra Kardan, Dan Stefan, Andrew Bysice, Samar Sayedyahosseini, Lina Dagnino, Matthew Cecchini, Sampath K Loganathan, Kathryn Roth, and Silvia Penuela
DOI: 10.1113/JP286172

Corresponding author(s): Silvia Penuela (silvia.penuela@schulich.uwo.ca)

The following individual(s) involved in review of this submission have agreed to reveal their identity: Maria Mayan (Referee #2)

Review Timeline:	Submission Date:	12-Mar-2024
	Editorial Decision:	12-May-2024
	Revision Received:	25-Sep-2024
	Accepted:	23-Oct-2024

Senior Editor: Peking Fong

Reviewing Editor: Jorge Contreras

Transaction Report:

Dear Dr Penuela,

Re: JP-RP-2024-286172 "Pannexin 1 and Pannexin 3 differentially regulate the tumorigenic properties of cutaneous squamous cell carcinoma" by Brooke L O'Donnell, Danielle Johnston, Ayushi Bhatt, Zahra Kardan, Dan Stefan, Andrew Bysice, Samar Sayedyahosseini, Lina Dagnino, Matthew Cecchini, Sampath K Loganathan, Kathryn Roth, and Silvia Penuela

Thank you for submitting your manuscript to The Journal of Physiology. It has been assessed by a Reviewing Editor and by 2 expert referees and we are pleased to tell you that it is potentially acceptable for publication following satisfactory major revision.

LANGUAGE EDITING AND SUPPORT FOR PUBLICATION: If you would like help with English language editing, or other article preparation support, Wiley Editing Services offers expert help, including English Language Editing, as well as translation, manuscript formatting, and figure formatting at www.wileyauthors.com/eoo/preparation. You can also find resources for Preparing Your Article for general guidance about writing and preparing your manuscript at www.wileyauthors.com/eoo/prepresources.

REVISION CHECKLIST:

Please upload two versions of your manuscript text: one with all relevant changes highlighted and one clean version with no changes tracked. The manuscript file should include all tables and figure legends, but each figure/graph should be uploaded as separate, high-resolution files. The journal is now integrated with Wiley's Image Checking service. For further details,

see: <https://www.wiley.com/en-us/network/publishing/research-publishing/trending-stories/upholding-image-integrity-wileys-image-screening-service>

We look forward to receiving your revised submission.

Yours sincerely,

Peying Fong
Senior Editor
The Journal of Physiology

REQUIRED ITEMS

- Author photo and profile. First or joint first authors are asked to provide a short biography (no more than 100 words for one author or 150 words in total for joint first authors) and a portrait photograph. These should be uploaded and clearly labelled together in a Word document with the revised version of the manuscript. See Information for Authors for further details.
- You must start the Methods section with a paragraph headed Ethical Approval. If experiments were conducted on humans, confirmation that informed consent was obtained, preferably in writing, that the studies conformed to the standards set by the latest revision of the Declaration of Helsinki and that the procedures were approved by a properly constituted ethics committee, which should be named, must be included in the article file. If the research study was registered (clause 35 of the Declaration of Helsinki), the registration database should be indicated, otherwise the lack of registration should be noted as an exception (e.g. The study conformed to the standards set by the Declaration of Helsinki, except for registration in a database). For further information see: <https://physoc.onlinelibrary.wiley.com/hub/human-experiments>.
- Please upload separate high-quality figure files via the submission form.
- You must upload original, uncropped western blot/gel images (including controls) if they are not included in the manuscript. This is to confirm that no inappropriate, unethical or misleading image manipulation has occurred. These should be uploaded as 'Supporting information for review process only'. Please label/highlight the original gels so that we can clearly see which sections/lanes have been used in the manuscript figures. For more information, see: <https://physoc.onlinelibrary.wiley.com/hub/journal-policies#imacmanip>.
- Please ensure that any tables are editable and in Word format, and wherever possible, embedded in the article file itself.
- Please ensure that the Article File you upload is a Word file.
- Your paper contains Supporting Information of a type that we no longer publish, including supplementary tables and figures. Any information essential to an understanding of the paper must be included as part of the main manuscript and figures. The only Supporting Information that we publish are video and audio, 3D structures, program codes and large data files. Your revised paper will be returned to you if it does not adhere to our Supporting Information Guidelines.

- Papers must comply with the Statistics Policy: https://jp.msubmit.net/cgi-bin/main.plex?form_type=display_requirements#statistics.

In summary:

- If $n \leq 30$, all data points must be plotted in the figure in a way that reveals their range and distribution. A bar graph with data points overlaid, a box and whisker plot or a violin plot (preferably with data points included) are acceptable formats.
- If $n > 30$, then the entire raw dataset must be made available either as supporting information, or hosted on a not-for-profit repository, e.g. FigShare, with access details provided in the manuscript.
- 'n' clearly defined (e.g. x cells from y slices in z animals) in the Methods. Authors should be mindful of pseudoreplication.
- All relevant 'n' values must be clearly stated in the main text, figures and tables.
- The most appropriate summary statistic (e.g. mean or median and standard deviation) must be used. Standard Error of the Mean (SEM) alone is not permitted.
- Exact p values must be stated. Authors must not use 'greater than' or 'less than'. Exact p values must be stated to three significant figures even when 'no statistical significance' is claimed.

- Please include an Abstract Figure file, as well as the Figure Legend text within the main article file. The Abstract Figure is a piece of artwork designed to give readers an immediate understanding of the research and should summarise the main conclusions. If possible, the image should be easily 'readable' from left to right or top to bottom. It should show the physiological relevance of the manuscript so readers can assess the importance and content of its findings. Abstract Figures should not merely recapitulate other figures in the manuscript. Please try to keep the diagram as simple as possible and without superfluous information that may distract from the main conclusion(s). Abstract Figures must be provided by authors no later than the revised manuscript stage and should be uploaded as a separate file during online submission labelled as File Type 'Abstract Figure'. Please also ensure that you include the figure legend in the main article file. All Abstract Figures should be created using BioRender. Authors should use The Journal's premium BioRender account to export high-resolution images. Details on how to use and access the premium account are included as part of this email.

EDITOR COMMENTS

Reviewing Editor:

Both reviewers highlighted the significance of this manuscript in the field of pannexin channels and its clinical relevance to skin cancer biology, and I concur with their assessments. Reviewer 1 raised several points that can be readily addressed and will enhance the prospects of this work. Reviewer 2 has two main concerns that require attention. Specifically, the suggestion to employ more specific tools for targeting Panx1 and Panx3, such as shRNA, in cell lines is feasible and would enhance the quality of this work.

Please also see 'Required Items' above.

Senior Editor:

Here, two Expert Referees and a Reviewing Editor offer detailed suggestions arising from their careful review of this initial version. Overall, I hope that you will find their points useful not only in guiding revision, but also in inspiring any future work. In preparing your revised manuscript, you are encouraged strongly to incorporate the experiments suggested by Referee 2, which the Reviewing Editor and I feel will improve this study profoundly.

In addition, The Journal's Statistics Policy states that data be expressed using means \pm standard deviation, rather than standard error of the mean, which I note persists within the legends of several figures (i.e. Figures 2 and 7). Similarly, the exact p values must be stated; adherence appears inconsistent. Please ensure full compliance.

Two Supplementary Figures are included. Per published policy on Supporting Information, if these figures are essential to understanding the study, then they must be incorporated in the manuscript proper. Please refer to https://jp.msubmit.net/cgi-bin/main.plex?form_type=display_requirements#suppinfo

Thank you for submitting your manuscript to The Journal of Physiology, and I look forward to receiving your revised manuscript.

REFeree COMMENTS

Referee #2:

The study "Pannexin 1 and Pannexin 3 differentially regulate the tumorigenic properties of cutaneous squamous cell carcinoma" investigates the role of pannexin 1 (PANX1) and pannexin 3 (PANX3) proteins in cutaneous squamous cell carcinoma (cSCC) tumours. While PANX1 levels are found to be increased in cSCC tumours and contribute to cell growth and migration, PANX3 levels are decreased and have a suppressive effect on the development of pre-cancerous lesions. This suggests that PANX1 has a pro-tumorigenic role, whereas PANX3 exhibits anti-tumorigenic properties in cSCC. The findings, derived from various models including culture, mouse, and patient-derived tissues, shed light on the mechanisms underlying keratinocyte malignant transformation and propose PANX1 and PANX3 as potential therapeutic targets for advanced cSCC treatment.

The paper is commendably well-written, offering clarity and coherence throughout, which greatly aids in comprehension and navigation of the complex subject matter. The integration of in vitro studies, animal experiments, and analysis of human samples provides a comprehensive perspective, enhancing the robustness and applicability of the findings. The novelty of the results, particularly regarding the contrasting roles of pannexin 1 and pannexin 3 in cutaneous squamous cell carcinoma (cSCC), holds significant promise for both the pannexin and cancer research communities. However, while the paper offers valuable insights, there are certain aspects that warrant further attention and elucidation to strengthen the overall impact and depth of the study.

Minor comments:

- Introduction section. The paragraph cited below seems to be more likely from the discussion section rather than the introduction. I suggest the authors to consider this comment and to move the paragraph or prepare a new one adapted to an introductory section.

"In a study using the 7,12-dimethylbenz(a)anthracene/12-Otetradecanoylphorbol-13-acetate (DMBA/TPA) mouse cutaneous carcinoma model, which mimics many aspects of human SCC (Abel et al., 2009), Panx3 was found genetically linked to body mass index and tumorigenesis by quantitative trait loci analysis in male but not female mice. It was also noted that pre-treatment Panx3 transcript levels in tail skin were positively associated with tumour susceptibility, where the higher the expression, the higher the risk of papilloma and carcinoma formation. Despite this, Panx3 transcript levels were markedly reduced in papillomas and carcinomas compared to untreated tail skin, suggesting that Panx3 may not influence tumour maintenance or progression (Halliwill et al., 2016). In human cSCC patient-derived tumours and

aged epidermis, immunofluorescence microscopy analysis using a small cohort of samples indicated that both PANX1 and PANX3 were visibly reduced in cSCC tumours (Cowan et al., 2012). However, data from the Human Protein Atlas shows conflicting results for PANX1 (PANX3 is not annotated), reporting that PANX1 is present at moderate levels in immunohistochemically stained cSCC tumour cores, where cells showed both membranous and cytoplasmic PANX1 localization. Collectively, these results suggest PANX1 and PANX3 levels are altered in cSCC tumorigenesis, but further investigation is needed to navigate the conflicting results and understand the effects of pannexin dysregulation"

- "Patient demographics" section should be included in the Methodology

Major comments:

- Figure 1A/B. Which cells were used to run the western-blot? Did the authors use human samples of different cell lines from cutaneous squamous cell carcinoma? Please indicate the type of cells and also if N=4 and N=3 refers to different biological samples or replicates in the case of cell lines.
- When analysing Panx1 protein localization (Figure 1C), it seems that the population of cells that naturally express Panx1 is much more consistent than the population of Panx1 overexpressed cells. Could the authors explain this fact? Panx1 levels are higher in cells that overexpress the protein, however only a few cells show this.
- Have the authors investigated the mechanism by which Panx1 blockers reduce cancer cells proliferation? (Figure 2A). Is it due to cell death or cell cycle arrest?
- Proliferation VS wound-healing assay. When proliferation was analysed (Figure 2A), SPL had a higher effect in proliferation than PBN. However in the wound healing experiments, similar effects were observed when using both inhibitors. Could the authors explain this fact? Is it related to the drug mechanism of action?
- When collecting human samples. Which were the selection criteria of the cohort?
- Human samples analysis. Immunohistochemistry techniques reveals that Panx1 is present not only in the tumoral area but also in the tumour microenvironment. Have the authors tested the specificity of the antibody? It seems that everything is positive. More magnification in the images would be appreciated. Furthermore, the authors should include a better explanation in the image to limiting each area within the same photo.
- Co-immunofluorescence of Panx1 and a tumoral marker would be much appreciated in the human samples.

Upon thorough review, it is evident that the paper exhibits substantial promise for publication. Should the authors diligently address and incorporate the suggested amendments provided in the comments, the manuscript stands for acceptance. The constructive feedback presented offers invaluable opportunities for refinement, enhancing the paper's clarity, coherence, and overall research contribution. With due diligence and attention to detail in addressing these comments, the paper is primed to meet the standards requisite for publication.

Referee #3:

In my opinion the major key points that have been explicitly stated by the authors in the article have not been adequately resolved by the experiments shown in the article.

1.- "In this study, using a combination of culture models, mouse models and patient-derived tissues, we found pannexin 1 levels are increased in cSCC tumours and present in all tumour cell types, functioning to increase cSCC cell growth and migration".

2.- "Taken together, our data indicates that in cSCC these pannexin family members seem to have opposite effects, where pannexin 1 is pro-tumorigenic and pannexin 3 is anti- tumorigenic".

Both sentences are too strong and not convincing with the data shown in this article.

Major points that should be address:

1.- Understanding the challenges inherent in studying both pannexins due to the absence of specific and selective channel blockers, as well as the scarcity of reliable antibodies for Panx3 protein, the authors of the study address these obstacles in their initial figures through diverse experiments targeting Panx1 and Panx3. For Panx1, they utilize the SCC-13 cell line

alongside non-selective Panx1 channel blockers, observing a reduction in cell migration and growth rate. However, due to the unavailability of Panx3 channel blockers, similar experiments cannot be conducted for Panx3. Nevertheless, given the capability demonstrated in Figure 1 to transfect cell lines, the authors could potentially incorporate experiments utilizing siRNAs or shRNAs to selectively diminish the expression of both Panx1 and Panx3, thus enabling a more direct examination of their respective effects within a comparable experimental model. Similarly, in the more compelling experiments showcased in Figure 3, wherein researchers employ Panx3 knockout (KO) mice to induce papilloma development at the base of the tail skin, the results strongly suggest that the absence of Panx3 enhances papilloma development and growth rate. To further elucidate these findings, a parallel experiment utilizing Panx1 KO mice could be conducted to investigate whether Panx1 KO yields contrasting results compared to Panx3 KO.

2.- The upregulation of Panx1 transcripts and protein in tumor tissue among patients presents compelling evidence. However, the results regarding Panx3 appear to be ambiguous. While Panx3 transcripts show a tendency towards reduction in tumor tissue, this trend is particularly pronounced in two out of eight patients (Figure 5B). Nonetheless, the protein levels detected via western blot present a puzzling scenario. The authors note an increase in Panx3 protein levels along with the presence of different-sized bands (32, 36, and 38 K) of unknown origin. It is imperative for the authors to clarify whether this reactive signal observed in the tumor patients' tissue corresponds to Panx3 fragments or not.

Minor comments and suggestions:

1.- "Immunocytochemistry using untransfected and PANX1- expressing SCC-13 cells (Fig. 1C) revealed that endogenous PANX1 showed a variety of presentations, where some cell clusters had a predominantly diffuse intracellular localization, whereas other clusters exhibited both cytosolic and prominent plasma membrane localization".

I think it is necessary to indicate the frequency of both types of localization forms.

2.- Figure 1: I don't believe that conducting experiments on transfection to overexpress Panx1 or Panx3 contributes significantly to the study. It may be advisable to either eliminate these experiments or provide additional explanations.

3.- Figure 2: The graphs in C and D do not display statistically significant symbols.

4.- In the discussion the authors indicate the following sentences that require tempering:

i.- "We also determined that in full thickness normal aged skin, PANX1 was present in low abundance whereas PANX3 levels were found at high levels".

It's advisable to avoid this sentence because the signals in western blots for Panx1 or Panx3 are detected using different antibodies, which could have varying affinities or titers.

ii.- "As for PANX3, we determined the protein localized predominantly intracellularly in SCC-13 cells, with a small pool trafficking to the cell surface, but we are unsure of the specific organelle compartments where PANX3 is present"

The authors should clarify that this applies specifically to the transfected Panx3-HA, as they were unable to observe endogenous Panx3 protein expression due to the lack of an antibody for immunofluorescence.

END OF COMMENTS

Confidential Review

12-Mar-2024

London, ON. Sept. 25th, 2024

To: Dr. Peking Fong and Dr. Jorge Contreras
Senior Editor and Guest Editor
Journal of Physiology

Dear Drs. Fong and Contreras,

Enclosed is our revised manuscript entitled: “**Pannexin 1 and Pannexin 3 differentially regulate the cancer cell properties of cutaneous squamous cell carcinoma**” authored by Brooke O’Donnell, Danielle Johnston, Ayushi Bhatt, Zahra Kardan, Dan Stefan, Andrew Bysice, Samar Sayedyahosseini, Lina Dagnino, Matthew Cecchini, Sampath Loganathan, Kathryn Roth and Silvia Penuela, to be considered for publication in *The Journal of Physiology* as part of the ‘Current advances in large-pore channels’ special issue.

We appreciate the positive reviews, the valuable comments from the editors and the timeline for revisions. We have done our best to address all the issues of the reviewers and we believe the paper is much stronger thanks to their suggestions. To improve the manuscript, we made changes to ensure the formatting, statistics and content follow journal standards, added new figure panels, an additional figure and a graphical abstract, and clarified the methods used. Please see below for a point-by-point response to reviewers’ comments.

This study is not under consideration for publication elsewhere, all authors have read and approved the submission of the manuscript and there are no conflicts of interest to declare.

Thank you for considering this revised manuscript and we look forward to hearing from you in the future.

Sincerely yours,

Silvia Penuela, Ph.D.
Associate Professor
Associate Chair - Research
Department of Anatomy and Cell Biology
Department of Oncology – Division of Experimental Oncology
Associate Scientist Lawson Research Institute
Schulich School of Medicine and Dentistry
Dental Science Building DSB 00061B
E-mail: spenuela@uwo.ca
Phone: [519-661-2111](tel:519-661-2111) ext. 84735 (office), 84445 (lab)
<https://www.schulich.uwo.ca/penuelalab/>

Re: MS #JP-RP-2024-286172

Title: Pannexin 1 and Pannexin 3 differentially regulate the cancer cell properties of cutaneous squamous cell carcinoma

REQUIRED ITEMS

- Author photo and profile. First or joint first authors are asked to provide a short biography (no more than 100 words for one author or 150 words in total for joint first authors) and a portrait photograph. These should be uploaded and clearly labelled together in a Word document with the revised version of the manuscript. See Information for Authors for further details.

We have uploaded the author profile text and image in the portal with the manuscript resubmission.

- You must start the Methods section with a paragraph headed Ethical Approval. If experiments were conducted on humans, confirmation that informed consent was obtained, preferably in writing, that the studies conformed to the standards set by the latest revision of the Declaration of Helsinki and that the procedures were approved by a properly constituted ethics committee, which should be named, must be included in the article file. If the research study was registered (clause 35 of the Declaration of Helsinki), the registration database should be indicated, otherwise the lack of registration should be noted as an exception (e.g. The study conformed to the standards set by the Declaration of Helsinki, except for registration in a database). For further information see: <https://physoc.onlinelibrary.wiley.com/hub/human-experiments>.

Our methods section begins with an Ethical Approval paragraph and we have updated it to include that the study conforms to the Declaration of Helsinki, but is not registered to the database.

- Please upload separate high-quality figure files via the submission form.

As requested, we have uploaded 600 dpi tiff files as well as .eps files for each figure via the submission form.

- You must upload original, uncropped western blot/gel images (including controls) if they are not included in the manuscript. This is to confirm that no inappropriate, unethical or misleading image manipulation has occurred. These should be uploaded as 'Supporting information for review process only'. Please label/highlight the original gels so that we can clearly see which sections/lanes have been used in the manuscript figures. For more information, see: <https://physoc.onlinelibrary.wiley.com/hub/journal-policies#imagmanip>.

Uncropped and labeled Western blots were uploaded as part of the ‘Supporting information for review process only’ in the submission portal.

- Please ensure that any tables are editable and in Word format, and wherever possible, embedded in the article file itself.

All tables are embedded in the manuscript revised text files and are editable.

- Please ensure that the Article File you upload is a Word file.

The manuscript article file uploaded is a .docx Microsoft Word file.

- Your paper contains Supporting Information of a type that we no longer publish, including supplementary tables and figures. Any information essential to an understanding of the paper must be included as part of the main manuscript and figures. The only Supporting Information that we publish are video and audio, 3D structures, program codes and large data files. Your revised paper will be returned to you if it does not adhere to our Supporting Information Guidelines.

We have moved Supplementary Figures 1 and 2 into the main text as Figure 4 and Figure 10, respectively.

- Papers must comply with the Statistics Policy: https://jp.msubmit.net/cgi-bin/main.plex?form_type=display_requirements#statistics.

In summary:

- If $n \leq 30$, all data points must be plotted in the figure in a way that reveals their range and distribution. A bar graph with data points overlaid, a box and whisker plot or a violin plot (preferably with data points included) are acceptable formats.

- If $n > 30$, then the entire raw dataset must be made available either as supporting information, or hosted on a not-for-profit repository, e.g. FigShare, with access details provided in the manuscript.

- 'n' clearly defined (e.g. x cells from y slices in z animals) in the Methods. Authors should be mindful of pseudoreplication.

- All relevant 'n' values must be clearly stated in the main text, figures and tables.

- The most appropriate summary statistic (e.g. mean or median and standard deviation) must be used. Standard Error of the Mean (SEM) alone is not permitted.

- Exact p values must be stated. Authors must not use 'greater than' or 'less than'. Exact p values must be stated to three significant figures even when 'no statistical significance' is claimed.

We have read the journal’s statistics policy as well as the points listed above and have made changes to ensure our data is in line with the requirements.

- Please include an Abstract Figure file, as well as the Figure Legend text within the main article file. The Abstract Figure is a piece of artwork designed to give readers an immediate understanding of the research and should summarise the main conclusions. If possible, the image should be easily 'readable' from left to right or top to bottom. It should show the physiological relevance of the manuscript so readers can assess the importance and content of its findings. Abstract Figures should not merely recapitulate other figures in the manuscript. Please try to keep the diagram as simple as possible and without superfluous information that may distract from the main conclusion(s). Abstract Figures must be provided by authors no later than the revised manuscript stage and should be uploaded as a separate file during online submission labelled as File Type 'Abstract Figure'. Please also ensure that you include the figure legend in the main article file. All Abstract Figures should be created using BioRender. Authors should use The Journal's premium BioRender account to export high-resolution images. Details on how to use and access the premium account are included as part of this email.

We have included an abstract figure file creating using our own premium BioRender account and the accompanying figure legend in the re-submission.

EDITOR COMMENTS

Reviewing Editor:

Both reviewers highlighted the significance of this manuscript in the field of pannexin channels and its clinical relevance to skin cancer biology, and I concur with their assessments. Reviewer 1 raised several points that can be readily addressed and will enhance the prospects of this work. Reviewer 2 has two main concerns that require attention. Specifically, the suggestion to employ more specific tools for targeting Panx1 and Panx3, such as shRNA, in cell lines is feasible and would enhance the quality of this work.

Senior Editor:

Here, two Expert Referees and a Reviewing Editor offer detailed suggestions arising from their careful review of this initial version. Overall, I hope that you will find their points useful not only in guiding revision, but also in inspiring any future work. In preparing your revised manuscript, you are encouraged strongly to incorporate the experiments suggested by Referee 2, which the Reviewing Editor and I feel will improve this study profoundly.

Thank you to the Reviewing and Senior Editors for the reviews and the opportunity to submit a revised version for further consideration. We have addressed your concerns as detailed below:

In addition, The Journal's Statistics Policy states that data be expressed using means +/- standard deviation, rather than standard error of the mean, which I note persists within the legends of

several figures (i.e. Figures 2 and 7). Similarly, the exact p values must be stated; adherence appears inconsistent. Please ensure full compliance.

We have adjusted all graphs to report means +/- standard deviation and ensured exact p values were included.

Two Supplementary Figures are included. Per published policy on Supporting Information, if these figures are essential to understanding the study, then they must be incorporated in the manuscript proper. Please refer to https://jp.msubmit.net/cgi-bin/main.plex?form_type=display_requirements#suppinfo

We have incorporated the Supplementary Figure 1 and Supplementary Figure 2 into the main manuscript text as Figure 4 and Figure 10, respectively.

Thank you for submitting your manuscript to The Journal of Physiology, and I look forward to receiving your revised manuscript.

Thank you.

REFEREE COMMENTS

Referee #2:

The study "Pannexin 1 and Pannexin 3 differentially regulate the tumorigenic properties of cutaneous squamous cell carcinoma" investigates the role of pannexin 1 (PANX1) and pannexin 3 (PANX3) proteins in cutaneous squamous cell carcinoma (cSCC) tumours. While PANX1 levels are found to be increased in cSCC tumours and contribute to cell growth and migration, PANX3 levels are decreased and have a suppressive effect on the development of pre-cancerous lesions. This suggests that PANX1 has a pro-tumorigenic role, whereas PANX3 exhibits anti-tumorigenic properties in cSCC. The findings, derived from various models including culture, mouse, and patient-derived tissues, shed light on the mechanisms underlying keratinocyte malignant transformation and propose PANX1 and PANX3 as potential therapeutic targets for advanced cSCC treatment.

The paper is commendably well-written, offering clarity and coherence throughout, which greatly aids in comprehension and navigation of the complex subject matter. The integration of in vitro studies, animal experiments, and analysis of human samples provides a comprehensive perspective, enhancing the robustness and applicability of the findings. The novelty of the results, particularly regarding the contrasting roles of pannexin 1 and pannexin 3 in cutaneous squamous cell carcinoma (cSCC), holds significant promise for both the pannexin and cancer research communities. However, while the paper offers valuable insights, there are certain

aspects that warrant further attention and elucidation to strengthen the overall impact and depth of the study.

We thank the reviewer for their enthusiasm for the work presented and their positive remarks. We have addressed your concerns as detailed below:

Minor comments:

- Introduction section. The paragraph cited below seems to be more likely from the discussion section rather than the introduction. I suggest the authors to consider this comment and to move the paragraph or prepare a new one adapted to an introductory section:

"In a study using the 7,12-dimethylbenz(a)anthracene/12-Otetradecanoylphorbol-13- acetate (DMBA/TPA) mouse cutaneous carcinoma model, which mimics many aspects of human SCC (Abel et al., 2009), Panx3 was found genetically linked to body mass index and tumorigenesis by quantitative trait loci analysis in male but not female mice. It was also noted that pre-treatment Panx3 transcript levels in tail skin were positively associated with tumour susceptibility, where the higher the expression, the higher the risk of papilloma and carcinoma formation. Despite this, Panx3 transcript levels were markedly reduced in papillomas and carcinomas compared to untreated tail skin, suggesting that Panx3 may not influence tumour maintenance or progression (Halliwill et al., 2016). In human cSCC patient-derived tumours and aged epidermis, immunofluorescence microscopy analysis using a small cohort of samples indicated that both PANX1 and PANX3 were visibly reduced in cSCC tumours (Cowan et al., 2012). However, data from the Human Protein Atlas shows conflicting results for PANX1(PANX3 is not annotated), reporting that PANX1 is present at moderate levels in immunohistochemically stained cSCC tumour cores, where cells showed both membranous and cytoplasmic PANX1 localization. Collectively, these results suggest PANX1 and PANX3 levels are altered in cSCC tumorigenesis, but further investigation is needed to navigate the conflicting results and understand the effects of pannexin dysregulation"

Thank you for the suggestion, we have adjusted the text, so it better reflects an introductory paragraph.

- "Patient demographics" section should be included in the Methodology

As suggested the patient demographics section was moved to the Methods section.

Major comments:

- Figure 1A/B. Which cells were used to run the western-blot? Did the authors use human samples of different cell lines from cutaneous squamous cell carcinoma? Please indicate the type of cells and also if N=4 and N=3 refers to different biological samples or replicates in the case of cell lines.

We have adjusted the figure legend to increase clarity of cell used and number of replicates.

- When analysing Panx1 protein localization (Figure 1C), it seems that the population of cells that naturally express Panx1 is much more consistent than the population of Panx1 overexpressed cells. Could the authors explain this fact? Panx1 levels are higher in cells that overexpress the protein, however only a few cells show this.

Transfections of both PANX1 and PANX3-HA plasmids into SCC-13 cells showed a very low efficiency of only 10%. Therefore, only a small percentage of cells exhibited pannexin overexpression. We have added a sentence regarding transfection efficiency to the corresponding results paragraph to explain this.

- Have the authors investigated the mechanism by which Panx1 blockers reduce cancer cells proliferation? (Figure 2A). Is it due to cell death or cell cycle arrest?

Figure 2A,B reported cell growth of SCC-13 cells in inhibitor and vehicle control conditions by analyzing cell numbers which is a sum of cell proliferation and death. Although we did not measure cell death directly, we performed an MTT assay to assess relative cell viability between conditions and determined that inhibitor concentrations of 1 mM for PBN and 20 μ M SPL showed no differences in cell viability compared to their corresponding vehicle control to ensure the effects observed were due to PANX1 channel blockade as opposed to inhibitor toxicity. Therefore, it is likely that treatment with PANX1 channel blockers triggers cell cycle arrest in SCC-13 cells. PBN and SPL treatment has been shown to reduce cell growth in other cancers such as melanoma (Freeman *et al.*, 2019 Cancers, Sayedyahosseini *et al.*, 2021 JBC), but mechanism of action of these compounds on inhibiting the cell cycle is currently unknown.

- Proliferation VS wound-healing assay. When proliferation was analysed (Figure 2A), SPL had a higher effect in proliferation than PBN. However in the wound healing experiments, similar effects were observed when using both inhibitors. Could the authors explain this fact? Is it related to the drug mechanism of action?

In both cell growth and migration assays the effect seen with SPL were evident earlier after initiating treatment and more pronounced than PBN, and we have updated panel D of Figure 2 to report significance differences at each time point to better display this difference. As stated in the discussion, we believe this difference is most likely due to the increased specificity and potency and decreased off-target effects when inhibiting PANX1 channels with SPL rather than PBN (Koval *et al.*, 2023, Curr Opin Pharmacol). Although we know SPL inhibition of PANX1 is direct and more specific than other inhibitors (Good *et al.*, 2018 Circ Rec), the exact mechanism of action of SPL inhibition of PANX1 is currently unknown.

- When collecting human samples. Which were the selection criteria of the cohort?

We have included information regarding patient selection criteria in the patient samples section of the methods.

- Human samples analysis. Immunohistochemistry techniques reveals that Panx1 is present not only in the tumoral area but also in the tumour microenvironment. Have the authors tested the specificity of the antibody? It seems that everything is positive. More magnification in the images would be appreciated. Furthermore, the authors should include a better explanation in the image to limiting each area within the same photo.

Thank you for the suggestion. The specificity of the antibody was confirmed by a secondary only control, which is commonplace in pathology when using DAB staining, as well as negative staining in acellular structures such as keratin pearls. For increased clarity, the supplementary figure outlining the antibody controls (Supplementary Figure 2) was moved into the main text (now Figure 9A and 10), and we included magnified secondary only control images which correspond to the magnified DAB images in Figure 9 (Figure 6 in original submission). We have also increased the magnification of the images in Figure 9 from 23x to 40x magnification and altered our figure legend explanation to better outline each area within the same image.

- Co-immunofluorescence of Panx1 and a tumoral marker would be much appreciated in the human samples.

For cSCC tumour diagnosis and carcinoma cell identification, pathologists use histology rather than immunohistochemical markers. To address your suggestion we had a board-certified pathologist annotate the cSCC tumours and have provided magnified images in Figure 9 which highlight different tumour regions including carcinoma cells, stromal cells, blood vessels and normal keratinocytes.

Upon thorough review, it is evident that the paper exhibits substantial promise for publication. Should the authors diligently address and incorporate the suggested amendments provided in the comments, the manuscript stands for acceptance. The constructive feedback presented offers invaluable opportunities for refinement, enhancing the paper's clarity, coherence, and overall research contribution. With due diligence and attention to detail in addressing these comments, the paper is primed to meet the standards requisite for publication.

Thank you for your thorough review. We have implemented the suggested amendments and agree that the manuscript is better for it.

Referee #3:

In my opinion the major key points that have been explicitly stated by the authors in the article have not been adequately resolved by the experiments shown in the article. Both sentences are too strong and not convincing with the data shown in this article.

1.- "In this study, using a combination of culture models, mouse models and patient-derived tissues, we found pannexin 1 levels are increased in cSCC tumours and present in all tumour cell types, functioning to increase cSCC cell growth and migration".

2.- "Taken together, our data indicates that in cSCC these pannexin family members seem to have opposite effects, where pannexin 1 is pro-tumorigenic and pannexin 3 is anti- tumorigenic".

We have adjusted the above statements to better reflect the data shown in this article.

Major points that should be address:

1.- Understanding the challenges inherent in studying both pannexins due to the absence of specific and selective channel blockers, as well as the scarcity of reliable antibodies for Panx3 protein, the authors of the study address these obstacles in their initial figures through diverse experiments targeting Panx1 and Panx3. For Panx1, they utilize the SCC-13 cell line alongside non-selective Panx1 channel blockers, observing a reduction in cell migration and growth rate. However, due to the unavailability of Panx3 channel blockers, similar experiments cannot be conducted for Panx3. Nevertheless, given the capability demonstrated in Figure 1 to transfect cell lines, the authors could potentially incorporate experiments utilizing siRNAs or shRNAs to selectively diminish the expression of both Panx1 and Panx3, thus enabling a more direct examination of their respective effects within a comparable experimental model. Similarly, in the more compelling experiments showcased in Figure 3, wherein researchers employ Panx3 knockout (KO) mice to induce papilloma development at the base of the tail skin, the results strongly suggest that the absence of Panx3 enhances papilloma development and growth rate. To further elucidate these findings, a parallel experiment utilizing Panx1 KO mice could be conducted to investigate whether Panx1 KO yields contrasting results compared to Panx3 KO.

Thank you for your suggestion. To address the concerns of inhibitor specificity to PANX1 we opted to perform CRISPR/Cas9 nucleofection to selectively delete *PANX1* in SCC-13 cells instead of knockdown of *PANX1* using shRNA constructs since SCC-13 cells show very low transfection efficiency. We assessed cell viability (Cytotox Red positive cells) and growth of mass cultures and a new figure (Figure 3) has been added for these findings which are consistent with the PANX1 inhibitor results. We found that with 90% reduction of PANX1 levels in mass cultures nucleofected with *PANX1* gRNA SCC-13 growth was severely reduced compared to control mass cultures, but did not show any differences in cell viability. However, we do not feel that subjecting *Panx1* KO mice to the DMBA/TPA

protocol would be an appropriate experiment given that we found PANX1 to be more of a cancer promoter. Therefore, ablating *Panx1* would decrease papilloma formation further when subjected to DMBA/TPA treatment and we would detect even less papillomas than the very small number of papillomas we were able to identify in WT mice.

As for PANX3, we performed CRISPR/Cas9 nucleofection of *PANX3* in the normal keratinocyte cell line N/TERT-1 since we found that PANX3 is present at higher levels in normal skin/keratinocytes compared to cSCC cancer cells/tumours. As seen in Response Figure 1 below, we were unable to produce any *PANX3* KO N/TERT-1 clones with single cell colony selection. This finding was consistent with previous work from our group where primary keratinocytes isolated from *Panx3* KO mice showed severe cell attachment deficits and could not adhere in culture (O'Donnell *et al.*, 2023 J Invest Dermatol). Next, we tried shRNA knockdown of PANX3 as a more moderate approach to reduce PANX3 levels, but were unable to produce any clones with PANX3 reductions compared to Scr controls (see Response Figure 2). We thank the reviewer for their praise for the use of the *Panx3* KO mice in the DMBA/TPA model and agree that the findings provide strong evidence for *Panx3* deletion enhancing papilloma development and growth. We feel that this data is sufficient to confirm the effects demonstrated are specific to PANX3 *in vivo*.

Response Figure 1. CRISPR/Cas9 deletion of *PANX3* in N/TERT-1 keratinocytes did not produce any *PANX3* knockout clones

Western blotting with commercial anti-PANX3 antibody was used to assess *PANX3* CRISPR/Cas9 deletion in N/TERT-1 cells. Clones either received *PANX3*-targeting guide RNA (CRISPR Clones) or scramble control (Scr Clones) guide RNA during the nucleofection protocol. No visible differences were seen in PANX3 in CRISPR or Scr clones. NT, non-treated cells. Protein sizes in kDa.

Response Figure 2. shRNA knockdown of PANX3 in N/TERT-1 keratinocytes was unsuccessful

N/TERT-1 cells were transfected with either a PANX3 shRNA knockdown (A-D, represents different *PANX3* transcript-targeting constructs) or scramble control (Scr) plasmid. Immunoblotting with a commercial anti-PANX3 antibody showed no differences in PANX3 between Scr and knockdown plasmid clones. NT, non-transfected cells. GAPDH as protein loading control, protein sizes in kDa.

2.- The upregulation of Panx1 transcripts and protein in tumor tissue among patients presents compelling evidence. However, the results regarding Panx3 appear to be ambiguous. While Panx3 transcripts show a tendency towards reduction in tumor tissue, this trend is particularly pronounced in two out of eight patients (Figure 5B). Nonetheless, the protein levels detected via western blot present a puzzling scenario. The authors note an increase in Panx3 protein levels along with the presence of different-sized bands (32, 36, and 38 K) of unknown origin. It is imperative for the authors to clarify whether this reactive signal observed in the tumor patients' tissue corresponds to Panx3 fragments or not.

We agree with the concerns expressed by the reviewer and have had issues with PANX3 antibodies in human tissues and cells. Since the specific epitope of the commercially available PANX3 polyclonal antibody (Invitrogen) is proprietary, and we are unable to perform peptide pre-adsorption assays to confirm the specificity of the lower immunoreactive species, we have removed panels C-G from Figure 5 and the corresponding results text.

Minor comments and suggestions:

1.- "Immunocytochemistry using untransfected and PANX1- expressing SCC-13 cells (Fig. 1C) revealed that endogenous PANX1 showed a variety of presentations, where some cell clusters had a predominantly diffuse intracellular localization, whereas other clusters exhibited both cytosolic and prominent plasma membrane localization".

I think it is necessary to indicate the frequency of both types of localization forms.

Thank you for the suggestion, we have inserted a statement to indicate the frequency of each localization profile in the results for Figure 1C.

2.- Figure 1: I don't believe that conducting experiments on transfection to overexpress Panx1 or Panx3 contributes significantly to the study. It may be advisable to either eliminate these experiments or provide additional explanations.

We understand the reviewer's rationale that transfection experiments may not be appropriate given all other experiments are performed using endogenous expression in cells and tissue. However, we included the PANX-HA-expressing cells specifically to try to visualize PANX3 localization in cells since there are no human PANX3 antibodies available which is optimized for immunocytochemistry. We also included PANX1 overexpression in SCC-13 cells to analyze whether exogenous PANX1 is retained more intracellularly or predominantly localizes to the plasma membrane. Intracellular localization has been reported in other cancer cells such as melanoma (Freeman *et al.*, 2019 *Cancers*, Sayedyahosseini *et al.*, 2021 *JBC*) and may be relevant in PANX1 function.

3.- Figure 2: The graphs in C and D do not display statistically significant symbols.

Figure 2C does not display any statistically significant symbols because the treatment p value was greater than 0.05. However, we have updated panel D of Figure 2 to report significance differences at each time point.

4.- In the discussion the authors indicate the following sentences that require tempering:

i.- "We also determined that in full thickness normal aged skin, PANX1 was present in low abundance whereas PANX3 levels were found at high levels".

It's advisable to avoid this sentence because the signals in western blots for Panx1 or Panx3 are detected using different antibodies, which could have varying affinities or titers.

We have adjusted the sentence to avoid the comparison between antibodies.

ii.- "As for PANX3, we determined the protein localized predominantly intracellularly in SCC-13 cells, with a small pool trafficking to the cell surface, but we are unsure of the specific organelle compartments where PANX3 is present"

The authors should clarify that this applies specifically to the transfected Panx3-HA, as they were unable to observe endogenous Panx3 protein expression due to the lack of an antibody for immunofluorescence.

Thank you for the suggestion, we have updated the text to reflect this point.

Dear Dr Penuela,

Re: JP-RP-2024-286172R1 "Pannexin 1 and Pannexin 3 differentially regulate the cancer cell properties of cutaneous squamous cell carcinoma" by Brooke L O'Donnell, Danielle Johnston, Ayushi Bhatt, Zahra Kardan, Dan Stefan, Andrew Bysice, Samar Sayedyahosseini, Lina Dagnino, Matthew Cecchini, Sampath K Loganathan, Kathryn Roth, and Silvia Penuela

We are pleased to tell you that your paper has been accepted for publication in The Journal of Physiology.

Yours sincerely,

Peying Fong
Senior Editor
The Journal of Physiology

If you would like to receive our 'Research Roundup', a monthly newsletter highlighting the cutting-edge research published in The Physiological Society's family of journals (The Journal of Physiology, Experimental Physiology, Physiological Reports, The Journal of Nutritional Physiology and The Journal of Precision Medicine: Health and Disease), please click this link, fill in your name and email address and select 'Research Roundup':

<https://www.physoc.org/journals-and-media/membernews>

- You can help your research get the attention it deserves! Check out Wiley's free Promotion Guide for best-practice recommendations for promoting your work at: www.wileyauthors.com/eoo/guide. You can learn more about Wiley Editing Services which offers professional video, design, and writing services to create shareable video abstracts, infographics, conference posters, lay summaries, and research news stories for your research at: www.wileyauthors.com/eoo/promotion.

The Corresponding Author will receive an email from Wiley with details on how to register or log-in to Wiley Authors Services where you will be able to place an order

EDITOR COMMENTS

Reviewing Editor:

I agree with the reviewers and am satisfied with the authors' responses.

Senior Editor:

Thank you for conscientiously responding to comments offered in the review of the original manuscript. Both Referees and the Reviewing Editor concur that this is a much-improved manuscript, and remark on its potential for advancing the collective knowledge base of pannexins and their roles in skin cancer.

We note that a few important, specific details pertaining to terminal animal (mouse) procedures remain missing. I understand that you do refer to an institution-approved animal protocol. Therefore, in the interest of transparency, we ask that you include details regarding euthanasia, including any method(s) of anesthesia, within the body of the manuscript before its final acceptance.

Many thanks for your understanding.

UPDATE FROM EDITORIAL OFFICE:

In light of Editor's comments above, and to save you some time, we have made a small amendment to your article file to improve transparency and compliance with our animal ethics policy. Specifically, we have moved the sentence regarding CO2 inhalation to the END of the paragraph in the 'DMBA/TPA mouse model' section of the Methods, and have added a note about using a rising concentration of CO2 (a journal requirement). The final sentence now reads: 'At endpoint, mice were sacrificed using CO2 inhalation (in a rising concentration).'

This same sentence has ALSO been added to the 'Ethical Approval' section of the Methods, so that this now reads: 'Animal experiments were approved by the Animal Care Committee at the University of Western Ontario (London, ON, Canada, Protocol #2022-025). At endpoint, mice were sacrificed using CO2 inhalation (in a rising concentration). For collection of human samples, the study was conducted in accordance with the latest version of the Declaration of Helsinki (except for registration in a database), and the protocol was approved by the Health Science Research Ethics Board (HSREB) of Western University and London Health Sciences Centre (London, ON, Canada, HSREB#103381). Subjects provided informed consent to participate in this study and tumour identity was determined by a pathologist.'

Please check your proofs very carefully in due course to make sure everything is OK.

REFeree COMMENTS

Referee #2:

Thank you very much for the revision. All the comments have been addressed and the paper is ready for publication.

Referee #3:

The authors have satisfactorily addressed my concerns and suggestions. Furthermore, I was convinced by the explanations in the response to the reviewers' letter. Therefore, I have no further questions. I think this article presents interesting results that will be useful in this field.